# MEDSCAMA: MEDICAL SCALE-AWARE MULTI-AGENT FRAMEWORK FOR MEDICAL IMAGE RETRIEVAL AND RETRIEVAL-AUGMENTED GENERATION

## ABSTRACT

Medical image retrieval and retrieval-augmented generation require representations that capture both visual similarity and clinically meaningful semantics across multiple levels of granularity. However, existing image encoders may miss pixel-to-organ-to-context cues, and report encoders may flatten hierarchical findings with global summaries, weakening vision-language alignment and alignment between query and targets in retrieval tasks. To address these gaps, we propose MedSCAMA (Medical SCale-Aware Multi-Agent Framework), a collaborative multi-agent framework designed to enable multi-scale feature modeling and retrieval-augmented reasoning. Specifically, ScaFormer encodes multi-scale visual features, the RU and PC Agents provide hierarchical text and similarity cues, and the QA Agent performs adaptive retrieval for evidence-grounded reasoning. Experiments across multiple medical imaging benchmarks demonstrate that MedSCAMA substantially enhances both retrieval quality and evidence-grounded answering, offering more accurate, interpretable, and clinically relevant results than existing approaches. This multi-scale, multi-agent design provides a principled foundation for integrating vision, language, and reasoning in medical AI systems.

## 1 INTRODUCTION

With the rapid expansion of imaging modalities such as X-ray, CT, MRI, and PET, hospitals now generate massive amounts of heterogeneous medical images. Medical image retrieval (MIR) thus plays a critical role in enabling clinicians to efficiently access large-scale imaging archives for diagnosis, treatment planning, education, and research. Recent advances in deep learning have significantly boosted MIR performance by comparing case-level similarities using semantic representations extracted from deep neural networks (Qayyum et al., 2017; Choe et al., 2022), and cross-modal embedding models such as MedCLIP (Wang et al., 2022) have further enabled joint image-text representation learning for MIR. Building upon these representation learning advances, retrieval-augmented generation (RAG) approaches have recently emerged as a useful framework for vision-language tasks in the medical domain (He et al., 2024; Wu et al., 2025; Zhang et al., 2024), which leverages retrieved evidence to improve the factual grounding and semantic alignment of multimodal representations in generation, and could directly benefit from MIR improvements.

However, existing medical image retrieval frameworks face two major limitations. At the representation level, single-scale visual encoders may conflate pixel-level lesions, regional structures, and whole-organ context into coarse image-level embeddings, while non-hierarchical textual encoders could fail to distinguish between entity-level findings and section-level summaries. At the retrieval stage, the features from different scales may distribute differently across cases, leading to unreliable correspondence and suboptimal matching. As illustrated in Figure 1, both limitations can lead to retrieval being dominated by visually salient but potentially irrelevant features, such as general contrast similarity or ECG leads. In contrast, clinicians can visually assess multi-scale features and align visual cues with multiple hierarchical levels in reports, enabling more flexible and discriminative matching that is less sensitive to such spurious cues.

Therefore, we propose MedSCAMA (Medical SCale-Aware Multi-Agent Framework), a novel framework that combines scale-aware multimodal backbone models with collaborative scale-

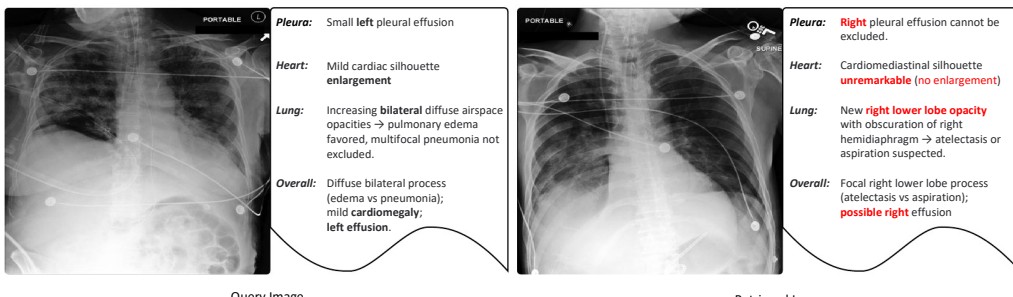

Figure 1: Failure case of current SOTA retrieval methods. Although the retrieved image closely resembles the query image in modality and anatomical region, the associated reports differ substantially. The image encoder (e.g., MedCLIP (Wang et al., 2022) used for this example) focused on salient but clinically irrelevant cues such as ECG leads, central venous catheters (CVC), or endotracheal tubes (ETT).

matching agents for medical image retrieval and retrieval-augmented generation. Our key contributions are:

- We design a ScaFormer with a scale-conditioned, sliding-window Mixture-of-Experts (ScaMoE) for multi-scale visual feature extraction. Neighboring image scales share experts through a sliding-window mechanism with ScaMoE, so adjacent resolutions route to overlapping expert sets.

- We introduce a Report Understanding (RU) agent and a Pairwise Comparison (PC) agent for hierarchical feature extraction and similarity labeling for text information. The RU agent decomposes clinical reports into a hierarchical text form, and the PC agent produces continuous signals for pairwise comparison similarity via an active learning process.

- We introduce a novel multi-scale feature alignment framework that comprises two coupled processes: soft contrastive learning that produces reward signals to update both the ScaFormer and the Pairwise Comparison (PC) agent, and direct preference optimization of the PC agent using text inputs processed by the RU agent.

- We develop a collaboration framework for RAG with our Question Answering (QA) agent and ScaFormer: the QA agent uses prompt engineering to produce a refined instruction for ScaFormer. ScaFormer then executes a progressive retrieval strategy under this instruction by selecting evidence from multi-scale databases, and sends the retrieved cases back to the QA agent for answer generation.

## 2 RELATED WORKS

**Medical Image Retrieval.** Recent advances in medical image retrieval increasingly exploit vision-language pre-training (VLP) from paired images and reports, moving beyond traditional supervised or metric/hashing approaches. For example, CXR-CLIP (You et al., 2023) expands image-label datasets into image-text pairs using prompt templates and multiple report sections, with study-level contrastive losses to improve chest X-ray image-report retrieval and zero/few-shot classification. Building upon the trend of leveraging textual reports, more recent work like RadIR (Zhang et al., 2025b) constructs large-scale benchmarks (MIMIC-IR for chest X-ray, CTRATE-IR for CT) and defines multi-grained similarity (from global image to specific anatomical regions) obtained from dense radiology reports, enabling image-image and image-report retrieval tasks with high fidelity. Despite these advancements in granularity, a common limitation persists: these methods typically learn a single, monolithic representation for an image or report. Consequently, they may conflate clinically distinct features that exist at different scales. Meanwhile, additional works (Ko & Park, 2025; Deanda et al., 2025) focus on handling clinical language aspects (e.g. negation, domain imbalance), and evaluating robustness of contrastive VLP models under distribution shifts or occlusions, further underscoring the need for more nuanced representations, which further motivates our scale-aware approach.

**Medical Retrieval-Augmented Generation.** Recent studies have advanced multimodal retrieval-augmented generation (RAG), improving the factuality and robustness of medical vision-language models. MedDr (He et al., 2024) integrates retrieval at inference time, enabling generalist medical VLMs to leverage external evidence across diverse domains such as radiology, pathology, and dermatology, improving VQA, report generation, and clinical decision-making support without retraining. FactMM-RAG (Sun et al., 2025) focuses on radiology report factuality by using RadGraph-based retrieval to provide entity- and relation-consistent evidence for text generation. RULE (Xia et al., 2024b) addresses reliability by adaptively controlling the number of retrieved contexts and applying preference fine-tuning to reduce noise and over-reliance on irrelevant evidence. MMedRAG (Xia et al., 2025) generalizes this approach with domain-aware retrieval selection, adaptive context calibration, and preference optimization, achieving cross-domain gains in factuality and robustness. However, these methods typically rely on single-scale or global representations for retrieval, which may lack the granularity needed to align fine-grained visual findings with hierarchical textual evidence. In contrast, our approach is intended to enhance the retrieval selection and content generation by introducing scale-aware architecture design and training.

## 3 MEDSCAMA

We propose MedSCAMA (Medical SCale-Aware Multi-Agent Framework), a unified framework for medical image understanding that integrates multi-scale representation learning, multi-agent collaboration, and retrieval-augmented reasoning, as illustrated in Figure 2. MedSCAMA combines the ScaFormer encoder with scale-aware Mixture-of-Experts (ScaMoE) for hierarchical feature extraction, an RU Agent for multi-scale text decomposition, a PC Agent for soft contrastive alignment with preference optimization, and a QA Agent for progressive retrieval and question answering. Together, these components enable scale-aware representation learning and retrieval-grounded reasoning, leading to more accurate and clinically meaningful medical image retrieval and analysis.

### 3.1 SCAFORMER

We build ScaFormer upon the Q-Former design from BLIP (Li et al., 2022), which bridges the image encoder and the LLM by extracting a fixed number of output features independent of image resolution. Specifically, we first process each image $I_i$ in $\mathcal{I} = \{I_i\}_{i=1}^N$ with a pretrained Swin Transformer (Liu et al., 2021) to obtain multi-scale image features $\{I_{i,s}\}_{s=1}^S$ where $N$ denotes the total number of samples and $S$ the number of scales.

Different from the standard Q-Former, we apply independent cross-attention modules for each input image scale as illustrated in Figure 2. Then, to better utilize information from multiple scales and exploit correlations across adjacent scales, we propose a scale-aware Mixture-of-Experts (ScaMoE) module with a sliding-window routing strategy.

For each feature $I_{i,s}^{attn}$ as the output of $I_{i,s}$ by the above scale-specific attention mechanism, the router $\nu$ assigns a masked adaptive weight vector to a set of experts $\mathcal{E} = \{E_0, E_1, \ldots, E_{s+w-1}\}$ and $E_0$ is the shared expert activated for all scales and $w$ denotes the window size of activation. For an optimal user prompt embedding $p_Q$, the ScaMoE module is defined as:

$$I_{i,s}^{moe} = \sum_{E \in \mathcal{E}} \nu_E([I_{i,s}^{attn}, p_Q]) E(I_{i,s}^{attn}), \quad \nu_E([I_{i,s}^{attn}, p_Q]) = \frac{\exp(g_E([I_{i,s}^{attn}, p_Q]))}{\sum_{E \in \mathcal{E}} \exp(g_E([I_{i,s}^{attn}, p_Q]))}, \quad (1)$$

where $[I_{i,s}^{attn}, p_Q]$ is the concatenation of feature $I_{i,s}^{attn}$ and prompt $p_Q$, and the gating function $g_E(\cdot)$ is implemented as

$$g_E([I_{i,s}^{attn}, p_Q]) = \begin{cases} \mathbf{w}_E^T \sigma(W[I_{i,s}^{attn}, p_Q] + b), & \text{index}(E) \in [s, s+w-1] \cup \{0\}, \\ -\infty, & \text{otherwise}, \end{cases} \quad (2)$$

where $W, b$ are learnable parameters, $\sigma(\cdot)$ is the activation function, and $\mathbf{w}_E$ projects the hidden state to the routing logit for expert $E$.

The gating mechanism uses a sliding window of size $w$ and enables neighboring scales to share partially overlapping experts. This design ensures differentiable routing weights while allowing each expert to specialize on scale-specific semantics. In our experiments, we set $w = 3$ and $S = 4$.

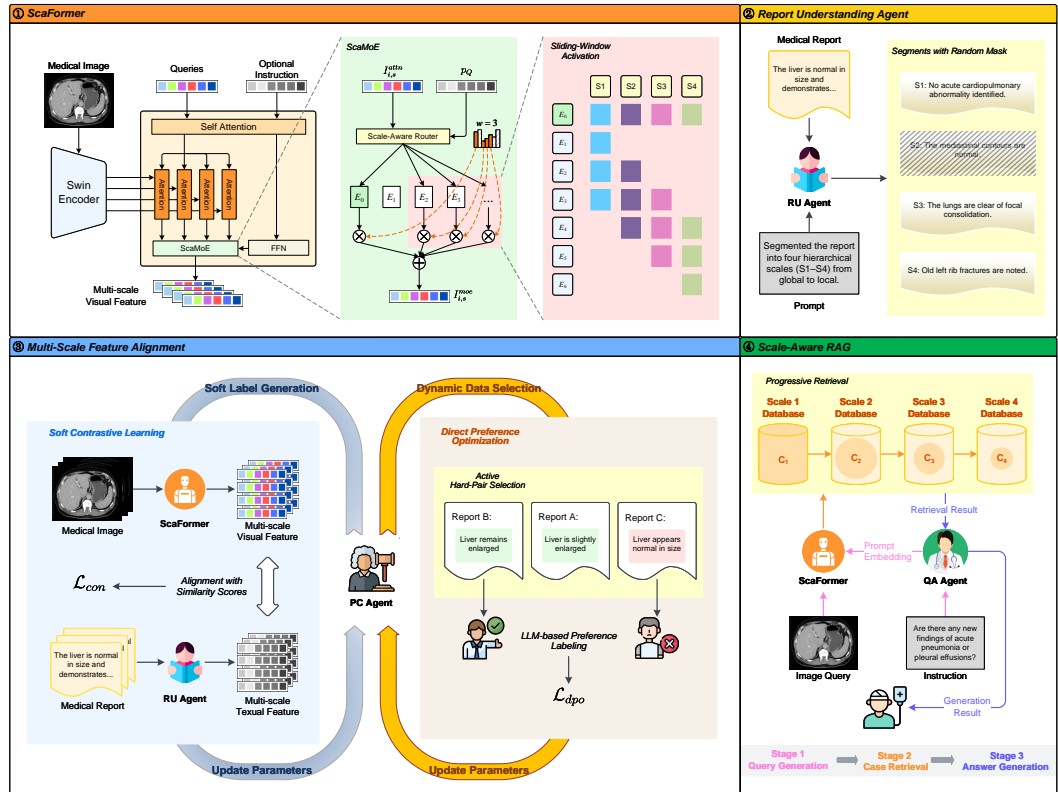

Figure 2: Overview of MedSCAMA. Our framework integrates multi-scale soft contrastive learning with the ScaFormer (shown in ①, with detaied ScaMoE and Sliding-Window Activation) and two LVLM agents: the RU Agent (②) for report understanding and the PC Agent for pairwise similarity scoring (③), providing continuous supervision signals. QA Agent (④) performs adaptive retrieval and auxiliary reasoning through three stages: Query Generation, Progressive Retrieval, and Answer Generation.

Finally, the ScaFormer module outputs the set of multi-scale features

$$\mathcal{V} = \{\mathbf{v}_{i,s} : \mathbf{v}_{i,s} = \text{ScaFormer}(I_{i,s}, p_Q) \text{ for } 1 \leq i \leq N, 1 \leq s \leq S\} \tag{3}$$

The features are subsequently employed in the multi-scale contrastive alignment loss (Algorithm 1) to jointly learn shared anatomical priors and fine-grained, scale-specific patterns for retrieval and reasoning tasks.

## 3.2 REPORT UNDERSTANDING AGENT SUPERVISED TUNING

To enable structured representation learning of medical reports, we fine-tune a Report Understanding (RU) Agent via supervised instruction tuning. Given a raw medical report $R_i$, we first construct a multi-scale segmentation dataset using GPT-5 as a teacher model. Each report is decomposed into four hierarchical scales from global summaries to local findings. Specifically, Scale 1 provides an overall conclusion for the entire report. Scale 2 summarizes observations at the whole-organ or system level, Scale 3 includes sentences about specific regions within an organ, and Scale 4 contains sentences describing very localized findings.

Formally, we model the RU Agent $\pi_{\text{RU}}$ as a multi-scale text generation network that produces hierarchical textual segments $\{R_{i,s}\}_{s=1}^S$ from the input report $R_i$ where each $R_{i,s}$ denotes the generated sentence of the original report $R_i$ at scale $s$. GPT-5 was prompted to produce ground-truth multi-scale annotations $\{R_{i,s}^*\}_{s=1}^S$ for each $R_i$, enabling supervised fine-tuning of the generative model as described in Algorithm 1.

---

**Algorithm 1:** Alternating Multi-Scale Contrastive Learning and DPO with Active Sampling

---

**Input:**

- Image set $\mathcal{I} = \{I_i\}_{i=1}^N$, Report set $\mathcal{R} = \{R_i\}_{i=1}^N$
- ScaFormer, PC agent $\pi_{PC}$, RU agent $\pi_{RU}$, Parameters $\Theta_{SF}, \Theta_{PC}$
- Temperature $\tau$, Max rounds $T_{\max}$

**Output:** Optimized parameters $\Theta_{SF}, \Theta_{PC}$

**for** $t = 1$ **to** $T_{\max}$ **do**

    **Step 1: Multi-Scale Feature Extraction**;

    **for** *each mini-batch* $\{(I_i, R_i)\}_{i=1}^B$ **do**

        1. Scale Decomposition: $\{(I_{i,s}, R_{i,s})\}_{1 \leq i \leq B, 1 \leq s \leq B} \leftarrow \{(I_i, R_i)\}_{i=1}^B$ by $\pi_{RU}$ and Swin-Transformer;

        2. Refined Multi-Scale Representation: $\mathcal{V} = \{\mathbf{v}_{i,s}\}_{i,s} \leftarrow \text{ScaFormer}(I_{i,s}|\Theta_{SF})$;

        3. Medical Knowledge Encoding: $\mathcal{T} = \{\mathbf{t}_{j,s}\}_{j,s} \leftarrow \text{MedEnc}(R_{j,s})$;

        4. Predict continuous similarity signals: $C = \{c_{ijs}\}_{i,j,s} \leftarrow \pi_{PC}(\mathcal{T}|\Theta_{PC})$;

    **Step 2: Multi-Scale Soft Contrastive Learning**;

    4. Compute contrastive loss $\mathcal{L}_{con}$ ;

    5. Update $\Theta_{SF}, \Theta_{PC}$ with $\nabla \mathcal{L}_{con}$;

    **Step 3: Active Hard-Pair Mining with DPO**;

    6. Dynamically select hard triplets within each mini-batch based on model uncertainty and scale disagreement;

    7. Annotate selected triplets with LLM-generated preference labels;

    8. Optimize PC Agent parameters $\Theta_{PC}$ using the DPO loss $\mathcal{L}_{dpo}$;

    **Step 4: Check Convergence**;

    9. If validation loss stops improving or max round $T_{\max}$ reached: **break**;

---

To enhance the model's ability to handle incomplete or noisy contexts, we adopt a masked scale generation strategy: for each report, a subset of scales is randomly masked by $\mathcal{M}_i$ and the model is required to reconstruct the missing scales $\mathcal{M}_i$ conditioned on the remaining ones:

$$\mathcal{L}_{RU} = -\sum_{i=1}^B \log \pi_{RU}(R_{i,\mathcal{M}_i}^* \mid R_i \setminus R_{i,\mathcal{M}_i}^*). \tag{4}$$

By forcing the RU Agent to infer masked scales from partial hierarchical information, this objective strengthens its capacity to capture inter-scale dependencies and to remain robust when certain levels of granularity are missing or corrupted.

With the pretrained RU Agent, the multi-scale segments $\{R_{i,s}\}$ are then encoded by a pretrained medical language model:

$$\mathcal{T} = \{\mathbf{t}_{i,s}\}_{i,s} \text{ with } \{\mathbf{t}_{i,s}\}_{s=1}^S = \text{MedEnc}(\text{RU}(R_i)), \tag{5}$$

where $\text{MedEnc}$ denotes a frozen domain-specific language model pretrained on large-scale medical corpora. This encoder provides semantically rich textual embeddings that preserve hierarchical clinical knowledge across different abstraction levels.

### 3.3 MULTI-SCALE FEATURE ALIGNMENT

To project the multi-scale textual segments $\mathcal{T}$ into the same latent space as the visual representations $\mathcal{V}$ across multiple scales, we employ a soft contrastive loss

$$\mathcal{L}_{con} = -\frac{1}{B^2 S} \sum_{s=1}^S \sum_{i=1}^B \sum_{j=1}^B c_{ijs} \log \frac{\exp(\langle \mathbf{v}_{i,s}, \mathbf{t}_{j,s} \rangle / \tau)}{\sum_{k=1}^B \exp(\langle \mathbf{v}_{i,s}, \mathbf{t}_{k,s} \rangle / \tau)} \tag{6}$$

where $\tau$ is the hyperparameter, and $c_{ijs}$ is the continuous similarity score generated by a Pairwise Comparison (PC) Agent. The PC Agent (parameterized by $\Theta_{PC}$) is trained based on the clinical relevance scores generated by a large language model, and it generates continuous similarity

scores $c_{ijs} = \pi_{\text{PC}}(R_{j,s}|R_{i,s}) \in [0,1]$ for each report pair $(R_{i,s}, R_{j,s})$ to guide the optimization of ScaFormer.

To further improve the discrimination ability of the PC Agent, we adopt an active learning strategy. Let $\pi_{\text{PC}}(R_B|R_A)$ denote the similarity score that the report $R_B$ matches the anchor $R_A$. Given the current parameter $\Theta_{\text{PC}}$, we dynamically sample hard triplets $(R_A, R_B, R_C)$ where $\pi_{\text{PC}}(R_B|R_A) > \pi_{\text{PC}}(R_C|R_A)$ is annotated via an LLM-based preference labeling framework. The PC Agent is then trained following the direct preference optimization (DPO) loss (Rafailov et al., 2023). For each labeled triplet $(R_A, R_B, R_C) \in \mathcal{H}$ with preference $R_B \succ R_C$ conditioned on $R_A$, the objective is

$$\mathcal{L}_{\text{dpo}} = -\frac{1}{|\mathcal{H}|} \sum_{(R_A,R_B,R_C)\in\mathcal{H}} \left[ \log \sigma \left( \beta \log \frac{\pi_{\text{PC}}(R_B|R_A, \Theta_{\text{PC}})}{\pi_{\text{PC}}(R_B|R_A, \Theta_{\text{Ref}})} - \beta \log \frac{\pi_{\text{PC}}(R_C|R_A, \Theta_{\text{PC}})}{\pi_{\text{PC}}(R_C|R_A, \Theta_{\text{Ref}})} \right) \right] \tag{7}$$

where $\sigma$ is the sigmoid function, $\Theta_{\text{Ref}}$ is the parameter of the reference model and $\beta > 0$ controls the strength of the policy update relative to the reference model. DPO, combined with soft-contrastive reward signals from ScaFormer, enables a joint text-vision training of the PC agent.

This training alternates between optimizing $\mathcal{L}_{con}$ for multi-scale representation learning and $\mathcal{L}_{dpo}$ for preference modeling. Specifically, in each iteration, Algorithm 1 first updates $\Theta_{\text{PC}}$ and $\Theta_{\text{SF}}$ via $\mathcal{L}_{con}$, then refines $\Theta_{\text{PC}}$ through $\mathcal{L}_{dpo}$. The overall objective can be expressed as

$$\min_{\Theta_{\text{PC}},\Theta_{\text{SF}}} \mathcal{L}_{con} + \lambda \min_{\Theta_{\text{PC}}} \mathcal{L}_{dpo}, \tag{8}$$

where $\lambda$ balances the two stages. This alternating optimization continues until convergence, yielding multi-scale embeddings with improved semantic alignment and retrieval accuracy.

### 3.4 SCALE-AWARE RETRIEVAL-AUGMENTED GENERATION

We couple a VLLM-based Question Answering (QA) Agent with the ScaFormer encoder to realize adaptive, evidence-grounded reasoning over large medical databases. The QA Agent performs dynamic prompt tuning to produce task-specific queries, guides multi-scale feature extraction, drives coarse-to-fine retrieval, and finally consumes the retrieved evidence for answer generation.

Given a VQA question $Q$, the QA Agent $\pi_{\text{QA}}$ first generates a tunable prompt embedding via a learnable prompt-tuning module,

$$p_Q = \text{Emb}_{\pi_{\text{QA}}}(Q), \tag{9}$$

where $\text{Emb}_{\pi_{\text{QA}}}$ denotes the token-level logits generated by the QA agent which are conditioned on the input question. These logits are then projected into a continuous embedding space, allowing the prompt representation $p_Q$ to adapt dynamically to the retrieval and reasoning task.

ScaFormer then takes as input this prompt $p_Q$ together with multi-scale image queries $I^q = \{I_s^q\}_{s=1}^S$ produced by the Swin-Transformer, yielding question-adaptive multi-scale features

$$\{\mathbf{v}_s^q\}_{s=1}^S = \text{ScaFormer}(I^q, p_Q), \tag{10}$$

so that clinically relevant scales are emphasized rather than relying on static, task-agnostic features.

For a image-report dataset $\mathcal{D} = \{(I_i, R_i)\}$, we first process all samples in $\mathcal{D}$ with $\pi_{\text{RU}}$ and Swin-Transformer, and create a group of multi-scale datasets $\{\mathcal{D}_s\}_{s=1}^S$ where $\mathcal{D}_s = \{(\mathbf{v}_{i,s}, \mathbf{t}_{i,s})\}$ for a specific scale $s$. We perform a progressive retrieval starting from $\mathcal{D}_1$ the coarsest scale to $\mathcal{D}_S$ the finest scale with a search scheme $K = [K_1, \cdots, K_S]$, where $\{K_s\}$ are a group of pre-set selection numbers in each scale such that $K_1 > K_2 > \cdots > K_S > 0$. Let $\mathcal{C}_1 = \{1, \cdots, |\mathcal{D}|\}$ be a index set. Specifically, at each scale $s$, all cases with index in $\mathcal{C}_s$ are ranked by the similarity score defined as $\text{sim}_s(i) = \langle \mathbf{v}_s^q, \mathbf{v}_{i,s} \rangle$, and create a top-$K_s$ index subset of $\mathcal{C}_s$ as $\mathcal{C}_{s+1} = \text{Index}\left( \text{Top}K_s(\{\text{sim}_s(i)\}_{i\in\mathcal{C}_s}) \right)$ for the next scale selection range.

Finally, the above recursive selection process returns a index subset $\mathcal{C}_{S+1}$ as the retrieval result for the input $I^q$ and question $Q$. According to the retrieval result, the final evidence set is created as $\mathcal{G}(I^q, Q) = \{(\mathbf{v}_{i,s}, \mathbf{t}_{i,s}) : i \in \mathcal{C}_{S+1}, 1 \leq s \leq S\}$. The QA Agent generates the answer $y$ conditioned on the question, the question-conditioned multi-scale features, and the retrieved evidence:

$y \sim \pi_{\text{QA}}(\cdot \mid I^q, \mathcal{G}(I^q, Q))$. Both QA agent and the ScaFormer are updated via the generation objective:

$$\mathcal{L}_{\text{QA}} = -\mathbb{E}_{I^q, Q} \log \pi_{\text{QA}}(y_{\text{label}} \mid I^q, \mathcal{G}(I^q, Q)). \tag{11}$$

Thus, retrieval remains a differentiable, task-driven component whose parameters (prompt and encoder) are optimized end-to-end through the QA supervision, aligning feature extraction and evidence use directly with clinical reasoning quality.

## 4 EXPERIMENTS

### 4.1 EXPERIMENT SETTINGS

**Implementation Details.** For the language components, the PC Agent is built upon the Vicuna-7B backbone (Chiang et al., 2023), while the RU Agent and QA Agent adopt LLaVA-Med-1.5 7B (Li et al., 2024) as its base model. For the visual branch, we employ the pretrained Swin-Base image encoder (Liu et al., 2021) as the visual backbone, where all of their four scales are used in ScaFormer, which also determines the window size (w=3) and number of experts (E=6). Our retrieval models are trained on the MIMIC-CXR (Johnson et al., 2019a) and CT-RATE (Hamamci et al., 2024) datasets. During contrastive training, the text encoder is initialized from BioClinicalBERT (Alsentzer et al., 2019). All models are optimized using the AdamW optimizer (Loshchilov & Hutter, 2019) with hyperparameters $\beta_1 = 0.9$, $\beta_2 = 0.999$, weight decay of 0.05, $\lambda = 1$, and an initial learning rate of $10^{-3}$. We adopt a batch size of 128, weight decay of $10^{-2}$, and train for 10 epochs on NVIDIA Tesla H100 GPUs.

**Baseline Methods.** For the medical image retrieval task, we compare our approach with several vision-language foundation models: BioMedCLIP (Zhang et al., 2025a), pre-trained on 15M biomedical image-text pairs; MedCLIP (Wang et al., 2022), trained on MIMIC-CXR (Johnson et al., 2019a) and CheXpert (Irvin et al., 2019); PMCCLIP (Lin et al., 2023), pre-trained on 1.6M biomedical image-caption pairs from PMC-OA; CT-CLIP, a CT image-language model trained on CT-RATE (Hamamci et al., 2024); and RADIR (Zhang et al., 2025b), which provides two versions trained on both MIMIC-CXR and CT-RATE datasets. For the RAG task, we compare against: decoding-based methods including Greedy Decoding, Beam Search (Sutskever et al., 2014), DoLa (Chuang et al., 2024), OPERA (Huang et al., 2024), and VCD (Leng et al., 2024); multimodal RAG frameworks such as MedDr (He et al., 2024), FactMM-RAG (Sun et al., 2025), RULE (Xia et al., 2024b) and MmedRAG (Xia et al., 2025).

**Evaluation Datasets.** For the medical image retrieval task, we follow the dataset construction protocol of RADIR (Zhang et al., 2025b) to build two benchmark datasets: MIMIC-IR, derived from the MIMIC-CXR database (Johnson et al., 2019a) for chest X-ray retrieval, and CTRATE-IR, constructed from the CT-RATE dataset (Hamamci et al., 2024) for chest CT retrieval. For the RAG task, we evaluate on two widely used medical vision-language datasets: MIMIC-CXR and IU-Xray (Demner-Fushman et al., 2015), covering both medical VQA and report generation scenarios. Following (Xia et al., 2024a), we construct VQA benchmarks by automatically generating question-answer pairs from medical reports using GPT-4 (Achiam et al., 2023), with all answers formatted as yes/no for standardized evaluation.

**Evaluation Metrics.** For the retrieval task, we adopt Recall@$k$ to assess whether relevant items appear within the top-$k$ predictions, and NDCG@$k$ to measure ranking quality by comparing the Discounted Cumulative Gain (DCG) of the predicted ranking against the ideal ranking. DCG is defined as $\sum_{i=1}^{k} \frac{\text{rel}_i}{\log_2(i+1)}$ and $\text{rel}_i$ is the ground-truth relevance score of the item at position $i$, and NDCG@$k$ is computed as the ratio of the DCG of the predicted ranking to that of the ideal ranking. For ranking, relevance is defined by the clinical semantic similarity between radiology reports. Specifically, the report paired with the query image serves as the ground-truth reference, and candidates are ranked by RaTEScore (Zhao et al., 2024) between their associated reports and this reference.

For the RAG task, we report Accuracy, F1 score, and AUROC for medical VQA evaluation, while BLEU (Papineni et al., 2002) (reported as the average of BLEU-1 to BLEU-4 scores to balance local

Table 1: Retrieval performance (%) of different methods. We report performance across three tasks: Image-to-Image, Image-to-Text, and Conditional Image-to-Image retrieval using Recall@$k$ and NDCG@$k$ metrics. The best and second-best results are highlighted with red and blue backgrounds.

| Method | Image → Image | | | | Image → Text | | | | Conditional Image → Image | | |
|---|---|---|---|---|---|---|---|---|---|---|---|
| | Recall@5 | Recall@10 | Recall@100 | NDCG@100 | Recall@5 | Recall@10 | Recall@100 | NDCG@100 | Recall@3 | Recall@5 | Recall@10 |
| **MIMIC-IR (Chest X-Ray)** | | | | | | | | | | | |
| MedCLIP | 3.05 | 4.77 | 18.93 | 10.70 | 0.19 | 0.28 | 3.77 | 9.27 | 12.58 | 19.67 | 30.46 |
| BioMedCLIP | 2.04 | 3.30 | 12.68 | 10.27 | | | | | 14.30 | 20.04 | 29.47 |
| PMC-CLIP | 2.20 | 3.58 | 12.03 | 10.06 | | | | | 14.37 | 21.54 | 32.94 |
| RadIR | 5.18 | 6.94 | 21.29 | 10.88 | 4.33 | 6.88 | 25.34 | 11.00 | 20.83 | 26.59 | 37.08 |
| MedSCAMA | 8.45 | 12.03 | 26.54 | 17.69 | 8.17 | 10.57 | 31.43 | 17.90 | 26.04 | 34.23 | 44.68 |
| **CTRATE-IR (Chest CT)** | | | | | | | | | | | |
| CT-CLIP | 19.43 | 28.76 | 68.13 | 79.96 | 5.05 | 8.19 | 39.92 | 79.45 | 43.85 | 54.44 | 67.09 |
| RadIR | 20.75 | 30.57 | 72.80 | 80.49 | 6.65 | 12.99 | 52.91 | 80.84 | 55.23 | 66.29 | 76.12 |
| MedSCAMA | 27.32 | 36.60 | 79.37 | 86.58 | 10.68 | 16.15 | 62.05 | 86.72 | 62.53 | 73.65 | 84.49 |

Table 2: Model performance (%) of different methods based on LLaVA-Med-1.5 on the medical VQA and report generation tasks. We report accuracy, F1 score, AUROC, as well as BLEU, ROUGE-L, and METEOR for report generation. The best and second-best results are highlighted with red and blue backgrounds.

| Models | VQA | | | | | | Report Generation | | | | | |
|---|---|---|---|---|---|---|---|---|---|---|---|---|
| | **IU-Xray** | | | **MIMIC-CXR** | | | **IU-Xray** | | | **MIMIC-CXR** | | |
| | ACC | F1 | AUC | ACC | F1 | AUC | BLEU | ROUGE-L | METEOR | BLEU | ROUGE-L | METEOR |
| LLaVA-Med-1.5 | 75.47 | 64.04 | 67.46 | 75.79 | 80.49 | 68.84 | 9.64 | 12.26 | 8.21 | 12.11 | 13.05 | 11.16 |
| w Greedy | 76.88 | 65.59 | 68.74 | 78.32 | 86.75 | 71.13 | 11.47 | 15.38 | 12.69 | 16.63 | 14.26 | 14.19 |
| w Beam Search | 76.91 | 66.06 | 68.77 | 81.56 | 86.36 | 73.79 | 12.10 | 16.21 | 13.17 | 16.97 | 14.74 | 14.43 |
| w DoLa | 78.00 | 66.75 | 72.19 | 81.35 | 85.73 | 72.78 | 11.79 | 15.82 | 12.72 | 17.11 | 14.89 | 14.81 |
| w OPERA | 70.59 | 61.54 | 63.22 | 69.34 | 76.66 | 62.74 | 10.66 | 14.70 | 12.01 | 15.40 | 12.52 | 13.72 |
| w VCD | 68.99 | 54.35 | 61.08 | 70.90 | 75.57 | 64.61 | 10.42 | 14.14 | 11.59 | 15.18 | 12.30 | 13.38 |
| w MedDr | 83.33 | 67.80 | 77.15 | 55.16 | 56.18 | 58.47 | 12.37 | 16.45 | 13.50 | 18.59 | 15.72 | 16.77 |
| w FactMM-RAG | 84.51 | 68.51 | 77.07 | 77.38 | 86.81 | 70.09 | 14.70 | 18.05 | 15.92 | 18.71 | 15.84 | 16.82 |
| w RULE | 87.84 | 78.00 | 85.78 | 83.92 | 87.49 | 83.44 | 27.53 | 23.16 | 27.99 | 18.61 | 15.96 | 17.42 |
| MMed-RAG | 89.54 | 80.72 | 87.13 | 83.57 | 88.49 | 85.08 | 31.38 | 25.59 | 32.43 | 23.25 | 12.34 | 20.47 |
| MedSCAMA | 89.83 | 81.55 | 91.33 | 86.66 | 90.61 | 88.79 | 34.01 | 28.51 | 36.88 | 25.64 | 17.26 | 24.90 |

and global n-gram consistency), ROUGE-L (Lin, 2004), and METEOR (Banerjee & Lavie, 2005) are employed to evaluate the quality of generated medical reports.

## 4.2 MAIN RESULTS

In this section, we present a comprehensive comparison with a wide range of baseline methods and open-source Med-LVLMs on both medical image retrieval and RAG tasks.

**Retrieval Performance.** As shown in Table 1, our proposed MedSCAMA framework consistently outperforms all baselines on both MIMIC-IR (Chest X-Ray) and CTRATE-IR (Chest CT) benchmarks across multiple retrieval settings. On MIMIC-IR, MedSCAMA achieves substantial gains in Image-to-Image, Image-to-Text, and Conditional Image-to-Image retrieval, with particularly strong improvements on top-$k$ metrics. For example, in the Image-to-Image setting, Recall@5 increases from 5.18% to 8.45%, and NDCG@100 from 10.88% to 17.69%, indicating more accurate ranking and relevance modeling. Similar trends hold on CTRATE-IR, where Conditional Image-to-Image retrieval sees Recall@5 rise from 66.29% to 73.65% and Recall@10 from 76.12% to 84.49%. These results demonstrate the effectiveness of our scale-aware multi-agent design for fine-grained cross-modal alignment and adaptive retrieval across imaging modalities.

**RAG Performance.** Table 2 summarizes the performance of different methods on medical VQA and report generation tasks across the IU-Xray and MIMIC-CXR benchmarks. We observe that our proposed MedSCAMA framework consistently achieves the best results across all metrics. For the VQA task, MedSCAMA attains the highest accuracy, F1 score, and AUROC on both datasets, surpassing the strongest baseline MMed-RAG by clear margins. For the report generation task, MedSCAMA also demonstrates superior performance on BLEU, ROUGE-L, and METEOR metrics. On IU-Xray, BLEU improves from 31.38 (MMed-RAG) to 34.01, and METEOR from 32.43 to 36.88, showing its ability to produce fluent and semantically accurate reports. Similar trends are

observed on MIMIC-CXR. These results highlight the effectiveness of the scale-aware multi-agent design in enhancing both reasoning accuracy for medical VQA and linguistic quality for report generation, leading to consistent gains across diverse datasets and evaluation metrics.

## 5 ANALYSIS

In this section, we conduct a comprehensive analysis of each component within the MedSCAMA framework to better understand the source of performance gains across retrieval, adaptive reasoning, and multi-scale representation learning. We present a series of analytical experiments, including ablation studies on retrieval training, expert activation analysis for the QA Agent, and the impact of adaptive retrieval strategies.

**Ablation Studies for Retrieval Training.** Table 3 reports the effect of adding each component independently to the QFormer baseline on MIMIC-IR and CTRATE-IR. Adding ScaFormer alone improves retrieval accuracy across both datasets, confirming the value of scale-aware feature extraction for capturing anatomical and pathological details. Ablation on the number of experts further validates that our design with six experts and a sliding window size of three achieves the optimal trade-off between specialization and parameter efficiency. Multiscale report decomposition further boosts performance by providing hierarchical textual representations, enabling finer image-text alignment. The largest single gain comes from the PC Agent with soft contrastive learning, which raises MIMIC-IR Recall@10 for Image-to-Image retrieval from 6.10% to 10.18%, and CTRATE-IR from 29.67% to 34.57%. Finally, combining all modules yields the best overall results, showing that each component contributes complementary benefits and that their integration leads to the most effective multi-scale retrieval system.

Table 3: Performance comparison (%) with different components for retrieval training mechanism. We report Recall@10 and Recall@100 for both Image-to-Image and Image-to-Text tasks. The best and second-best results in each column (excluding the final row) are highlighted.

| Method | MIMIC-IR | | | | CTRATE-IR | | | |
|---|---|---|---|---|---|---|---|---|
| | Image → Image | | Image → Text | | Image → Image | | Image → Text | |
| | Recall@10 | Recall@100 | Recall@10 | Recall@100 | Recall@10 | Recall@100 | Recall@10 | Recall@100 |
| Q-Former | 6.10 | 20.79 | 6.53 | 26.58 | 29.67 | 70.34 | 12.31 | 50.19 |
| + Independent Cross-Attention | 6.37 | 20.83 | 6.69 | 26.91 | 28.73 | 70.80 | 13.11 | 52.33 |
| + MoE (no window) | 6.93 | 21.19 | 7.36 | 26.87 | 29.51 | 71.23 | 13.58 | 52.66 |
| + ScaFormer | 7.89 | 22.32 | 8.41 | 28.35 | 30.08 | 71.95 | 14.26 | 55.51 |
| + Multiscale Decomposition | 8.03 | 22.84 | 8.96 | 28.09 | 32.61 | 74.49 | 14.60 | 55.84 |
| + Soft Contrastive Learning | 10.18 | 24.53 | 9.47 | 29.32 | 34.57 | 76.37 | 16.03 | 59.10 |
| All Combined | 12.03 | 26.54 | 10.57 | 31.43 | 36.6 | 79.37 | 16.15 | 62.05 |

**How Effective is Multi-Scale Strategy for QA Agent?** We analyze the mean activation of experts in the ScaMoE module for both large-scale and small-scale questions from the VQA tasks (Figure 3). The visualization shows that large-scale questions lead to consistently higher activation in large-scale experts, while small-scale questions primarily activate experts specialized for finer-scale representations. This alignment between question granularity and expert activation validates that our training successfully learns scale-specific routing, ensuring that the QA Agent correctly attends to the appropriate scale during reasoning. Moreover, Table 4 demonstrates that incorporating adaptive retrieval further improves QA performance. Compared to the vanilla QA Agent + ScaFormer, the adaptive retrieval mechanism boosts VQA AUC from 88.37% to 91.33% on IU-Xray and from 86.74% to 88.79% on MIMIC-CXR. For report generation, BLEU scores also increase from 33.33 to 34.01 on IU-Xray and from 24.97 to 25.64 on MIMIC-CXR. These results confirm that the multi-scale strategy, together with adaptive retrieval, substantially enhances the reasoning capability of the QA Agent.

## 6 CONCLUSION

In this work, we presented MedSCAMA (Medical SCale-Aware Multi-Agent Framework), a unified framework for multi-scale medical image retrieval and retrieval-augmented generation. Our design

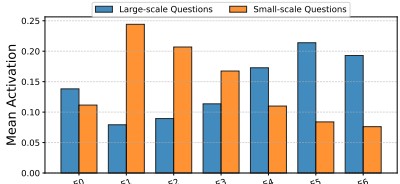

Figure 3: Mean activation of experts for ScaMoE.

Table 4: Performance comparison for QA agent under different modes.

| Models | VQA (AUC) | | Report Generation (BLEU) | |
|---|---|---|---|---|
| | IU-Xray | MIMIC-CXR | IU-Xray | MIMIC-CXR |
| QA Agent + ScaFormer | 88.37 | 86.74 | 33.33 | 24.97 |
| w Adaptive Retrieval | 91.33 | 88.79 | 34.01 | 25.64 |

integrates the ScaFormer network with a scale-sensitive Mixture-of-Experts (ScaMoE) to capture hierarchical visual semantics, an RU Agent for multi-scale textual decomposition, a PC Agent trained with soft contrastive learning for robust cross-modal alignment, and a QA Agent for adaptive evidence retrieval and evidence-grounded answering. Extensive experiments on both retrieval and RAG tasks demonstrate that MedSCAMA achieves state-of-the-art performance across multiple medical benchmarks. Beyond improving retrieval accuracy, our multi-agent, scale-aware design provides a principled approach for integrating vision, language, and reasoning in medical AI, paving the way for more interpretable and clinically useful decision support systems. While the framework demonstrates strong performance, its limitations include occasional retrieval misses in fine-grained matching and a dependency on the QA agent's reasoning ability, which we aim to address in future work by exploring hybrid retrieval strategies and enhanced fine-tuning.

ETHICS STATEMENT

This work uses only publicly released, de-identified medical datasets under their respective data-use agreements. For chest radiographs, we use MIMIC-CXR, which is explicitly de-identified to meet HIPAA Safe Harbor requirements and requires credentialed access via a data use agreement; we adhered to all access and usage restrictions (Johnson et al., 2019a). In compliance with the PhysioNet responsible-use policy for LLMs, any processing of MIMIC-CXR report text was conducted exclusively via the Azure OpenAI Service within a locked, enterprise-compliant environment that guarantees no data retention, training, or external sharing. For chest CT, we use CT-RATE, a paired CT-report dataset intended for research use (Hamamci et al., 2024). No new data collection or human-subject intervention was performed. Following the ICLR Author Guide and Code of Ethics, we include this statement to address potential concerns around privacy, consent, fairness, and safety in medical-AI research.

REPRODUCIBILITY STATEMENT

We provide full details to ensure reproducibility of our work: model architecture, training setups, and dataset preprocessing are detailed described, with hyperparameters, algorithms, and additional analyses included in the appendix. All datasets are publicly available under data use agreements, and we will release anonymized source code with scripts for training, inference, and evaluation in the supplementary materials. Together, these resources allow independent verification of all results and analyses reported in this paper.

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

# A  APPENDIX

## A.1  ACTIVE LEARNING FOR HARD TRIPLET MINING

At training round $t$, we leverage the previous PC Agent $\pi_{\text{PC}}(\cdot|\Theta_{\text{PC}}^{(t-1)})$ to actively identify challenging triplets for preference annotation. The goal is to efficiently focus labeling resources on *hard cases* that can maximally improve the ranking quality of the PC Agent in subsequent updates. The procedure consists of four steps: candidate generation, hard triplet selection, acquisition scoring, and preference-based optimization.

**Candidate Generation via Efficient Retrieval.**  For each anchor report $R_{i,s} \in \mathcal{U}$ from the unlabeled pool $\mathcal{U}$, we compute per-scale similarity scores:

$$c_{ijs}^{(t-1)} = \pi_{\text{PC}}(\mathbf{t}_{j,s}|\mathbf{t}_{i,s}, \Theta_{\text{PC}}^{(t-1)}), \tag{12}$$

where $\mathbf{t}_{i,s}$ denotes the textual embedding of report $R_i$ at scale $s$. We then average over scales to obtain a global similarity

$$c_{ij}^{(t-1)} = \frac{1}{S} \sum_{s=1}^{S} c_{ijs}^{(t-1)}. \tag{13}$$

To ensure scalability, we pre-filter candidates by retrieving the top-$K$ nearest neighbors $\mathcal{P}_i$ of $R_i$ in the MedEnc embedding space using an ANN index such as FAISS (Johnson et al., 2019b). Importantly, this embedding space is computed by a frozen medical text encoder and remains independent of the PC Agent training process, so that the pre-filtering stage provides a stable and unbiased candidate pool for subsequent preference-based hard triplet mining.

**Hard Triplet Construction.**  Following the standard hard/semi-hard triplet mining paradigm (Schroff et al., 2015), within each candidate set $\mathcal{P}_i$, we identify a *near-positive* report $R_B$ with the highest similarity score,

$$R_B = \arg\max_{j \in \mathcal{P}_i} c_{ij}^{(t-1)}, \tag{14}$$

and a near-negative report $R_C$ with the highest similarity among those constrained to be textually dissimilar to $R_i$:

$$R_C = \arg\max_{k \in \bar{\mathcal{P}}_i} c_{ik}^{(t-1)}, \quad \bar{\mathcal{P}}_i = \{k \in \mathcal{C}_i \mid \cos(\mathbf{t}_i, \mathbf{t}_k) < \delta_{\text{neg}}\}. \tag{15}$$

This ensures the negative report $R_C$ is semantically distinct yet confusing, forming a near-miss negative that the current PC Agent struggles to separate from the positive report $R_B$.

**Acquisition Scoring with Margin and Disagreement.**  Inspired by margin-based and disagreement-based acquisition strategies in active learning (Gal et al., 2017), to prioritize the most informative triplets, we define two criteria:

$$m_i = c_{iB}^{(t-1)} - c_{iC}^{(t-1)}, \qquad u_i = \text{Var}_s(c_{iB,s}^{(t-1)}) + \text{Var}_s(c_{iC,s}^{(t-1)}), \tag{16}$$

where $m_i$ measures the confidence gap between positive and negative samples, while $u_i$ captures disagreement across scales. The overall acquisition score is:

$$a_i = (1 - m_i) + \lambda \, u_i, \tag{17}$$

with $\lambda$ balancing the two terms. Here we use $1 - m_i$ instead of $-m_i$ for two reasons: (i) to ensure the score remains non-negative, since negative-valued indicators complicate hyperparameter tuning and weighting across terms, and (ii) to make the score more intuitive and directly comparable, so that higher values consistently correspond to more informative triplets after normalization and ranking.

After computing $a_i$ for all anchors $R_i$ in the unlabeled pool, we rank candidate triplets $\{(R_i, R_B, R_C)\}$ in descending order of $a_i$. Intuitively, a high $a_i$ indicates that the PC Agent is both *uncertain* (small margin $m_i$) and *inconsistent across scales* (large disagreement $u_i$) about the relative similarity of $R_B$ and $R_C$. Such triplets are expected to provide the greatest information gain if annotated.

**Discussion.** This active learning design ensures that annotation effort is concentrated on (i) hard triplets with high model uncertainty, (ii) disagreement cases that expose inconsistencies across scales, and (iii) near-miss negatives that are semantically similar yet clinically distinct. By iteratively refining the PC Agent with these informative samples, the model learns sharper decision boundaries and more reliable cross-scale preferences, ultimately improving retrieval quality with fewer labeled pairs.

## A.2 DATA STATISTICS

### A.2.1 DATASETS FOR RETRIEVAL TRAINING

We conduct all retrieval experiments on two large-scale medical vision-language datasets: MIMIC-CXR (Johnson et al., 2019a) and CT-RATE (Hamamci et al., 2024). The MIMIC-CXR dataset contains 377,110 chest X-ray images paired with corresponding radiology reports, while the CT-RATE dataset provides 25,692 chest CT images and reports curated from multiple institutions. Together, these datasets cover diverse imaging modalities and clinical findings, providing a comprehensive benchmark for evaluating cross-modal medical retrieval systems.

Following the preprocessing protocol established by RadIR (Zhang et al., 2025b), we standardize all image-report pairs by (i) removing incomplete studies without paired reports, (ii) normalizing image intensities across modalities, and (iii) converting free-text reports into structured sentence-level annotations using the RU Agent. For MIMIC-CXR, we adopt the official patient-wise train/validation/test splits to avoid patient overlap across subsets. For CT-RATE, we follow the same strategy as RadIR to ensure fair comparison across methods.

To make our results comparable with prior retrieval baselines, we replicate RadIR's pre-processing and evaluation pipeline. Specifically, both datasets are processed into multi-scale representations, where each report is segmented into hierarchical levels of abstraction, and each image is resized to multiple scales for visual feature extraction. This ensures our retrieval system and baselines are trained under identical conditions, isolating the contribution of our proposed methods rather than dataset differences.

Table 5: Statistics of medical image-report datasets used in our experiments.

| Dataset | # Image-Report Pairs |
|---|---|
| MIMIC-CXR | 377,110 |
| CT-RATE | 25,692 |

### A.2.2 DATASETS FOR RAG TASKS

We evaluate the retrieval-augmented generation (RAG) framework on two widely used medical vision-language datasets: IU-Xray (Demner-Fushman et al., 2015) and MIMIC-CXR (Johnson et al., 2019a). As summarized in Table 6, IU-Xray contains 589 images paired with 2,573 question-answer (QA) items, while MIMIC-CXR consists of 700 images and 3,470 QA pairs. Both datasets cover diverse radiological findings and diagnostic contexts, providing a comprehensive benchmark for evaluating question answering and report generation in medical imaging.

Following prior work (Xia et al., 2024b), we conduct visual question answering (VQA) evaluation using the question-answer pairs and data splits released by He et al. (2024), which were automatically generated from radiology reports and subsequently manually filtered to remove ambiguous or multi-image dependent cases. Each image is associated with multiple QA pairs covering local findings, organ-level attributes, and global diagnostic summaries. Answers are standardized into binary *Yes/No* or short categorical responses to facilitate quantitative evaluation using metrics such as Accuracy, F1 score, and AUROC.

For the report generation setting, we follow the same image-report pairs as in the original datasets. Each image serves as input to the model, which is required to produce a radiology report matching the reference ground truth. Evaluation metrics include BLEU (Papineni et al., 2002), ROUGE-L (Lin, 2004), and METEOR (Banerjee & Lavie, 2005), measuring both lexical overlap and semantic fidelity between generated and reference reports.

Table 6: Data statistics for RAG experiments on IU-Xray and MIMIC-CXR datasets.

| Dataset | IU-Xray | MIMIC-CXR |
|---|---|---|
| # Images | 589 | 700 |
| # QA Items | 2573 | 3470 |

## A.3 SUMMARIZATION OF BASELINES

### A.3.1 BASELINES FOR MEDICAL IMAGE RETRIEVAL

We evaluate SCAMA against several state-of-the-art vision-language foundation models that are widely used for medical image retrieval. Below we detail each baseline's architecture, training data, and strengths, to highlight where our contributions differ.

- **BioMedCLIP** (Zhang et al., 2025a): Pretrained on 15 million biomedical image-text pairs (PMC-15M), BioMedCLIP adapts the CLIP architecture for the biomedical domain by incorporating modality-specific augmentations and domain regularization. It demonstrates strong zero-shot retrieval performance on unseen disease categories, especially in image-to-text retrieval. However, it uses global contrastive objectives that treat all image-text pairs equally, which can limit its sensitivity to fine-grained local findings.

- **MedCLIP** (Wang et al., 2022): Trained on large chest X-ray datasets (MIMIC-CXR (Johnson et al., 2019a) and CheXpert (Irvin et al., 2019)), MedCLIP uses a decoupled image-text contrastive loss that seeks to reduce the effect of false negatives by reweighting or masking them, thereby improving retrieval robustness. It is particularly strong in recognizing common pathologies and extracting coarse anatomical similarities but shows degradation when distinguishing subtle lesion details or scale variance.

- **PMC-CLIP** (Lin et al., 2023): This model leverages 1.6 million biomedical figure-caption pairs from PMC-OA, many of which are subfigures or region-level captions. Because of this, PMC-CLIP learns alignments not only at full-image level but also partially at sub-regional or caption granularity. This gives PMC-CLIP an advantage over baselines when the retrieval query refers to localized or specific features, but its image encoder and caption processing still lack explicit multi-scale expert routing.

- **CT-CLIP** (Hamamci et al., 2024): Tailored for chest CT modalities, CT-CLIP is trained on CT-RATE, which includes volumetric and slice-based CT scans. Its encoder is adapted for 3D or volumetric feature extraction, providing better representation for CT's depth and tissue contrast. Nonetheless, CT-CLIP still primarily uses monolithic contrastive objectives without hierarchical decomposition of image or report, hence less sensitive to variations in scale within slices or across anatomical regions.

- **RADIR** (Zhang et al., 2025b): RADIR is a retrieval benchmark and modeling framework that defines similarity via radiology report content, supporting both whole-study and anatomy-conditioned image retrieval. It standardizes preprocessing (e.g., exclusion of incomplete pairs, consistent resizing, report cleaning) and provides splits on both MIMIC-CXR and CT-RATE datasets, enabling fair comparisons. Its retrieval models perform well on coarse image-to-image and image-to-text tasks, but they usually rely on global report embeddings and do not explicitly enforce scale-aware alignment between local features in images and report segments.

### A.3.2 BASELINES FOR RAG IN MED-LVLMS

We compare SCAMA with two major families of hallucination mitigation methods for medical LVLMs, including decoding-based methods that operate purely at the generation stage and multimodal retrieval-augmented methods that explicitly ground the model's outputs in external evidence:

- **Greedy Decoding** is a deterministic decoding strategy that selects the highest-probability token at each generation step. It is computationally efficient and widely adopted in real-time medical reporting systems due to its low latency. However, greedy decoding can

easily converge to locally optimal but globally suboptimal solutions, producing repetitive or truncated outputs and failing to capture clinically nuanced reasoning chains.

- **Beam Search (Sutskever et al., 2014)** is an extension of greedy decoding that maintains multiple candidate sequences (beams) throughout generation. By exploring the top-$k$ sequences at each step and selecting the highest-scoring one at the end, beam search improves the lexical diversity and semantic coherence of generated reports. In medical LVLMs, this broader search space helps avoid over-simplified interpretations and supports more comprehensive clinical descriptions.

- **DoLa (Chuang et al., 2024)** introduces layer-wise logit contrast to exploit differences between early- and late-layer representations in transformer-based LVLMs. Since factual knowledge often emerges in specific intermediate layers, DoLa encourages the model to align generation probabilities with the most informative layers, reducing spurious correlations that lead to hallucinated clinical findings.

- **OPERA (Huang et al., 2024)** adds an Over-trust Penalty and a Retrospection-Allocation mechanism to recalibrate attention distributions when LVLMs over-focus on summary tokens at the expense of visual evidence. In medical imaging, where fine-grained spatial details are critical for diagnosis, OPERA ensures that attention is allocated back to clinically meaningful image regions, thereby mitigating content hallucination.

- **VCD (Leng et al., 2024)** perturbs visual inputs and contrasts output distributions from the original and perturbed images to regularize model confidence. By enforcing consistency under visual distortions, VCD reduces over-reliance on unimodal language priors and compels the model to ground its predictions in authentic visual signals, leading to more reliable medical reasoning.

- **MedDr (He et al., 2024)** is a healthcare foundation model built on large-scale, diagnosis-focused datasets that integrates retrieval during inference. Given a medical image, MedDr retrieves clinically relevant reports or evidence passages and conditions the LVLM on them before generation, improving factual grounding and clinical decision support.

- **FactMM-RAG (Sun et al., 2025)** proposes a fact-aware multimodal retrieval-augmented pipeline that leverages RadGraph-based annotations to mine semantically aligned image-text pairs. The resulting retriever supplies high-quality evidence to guide the LVLM during radiology report generation, enhancing factual consistency across diverse pathologies and imaging modalities.

- **RULE (Xia et al., 2024b)** is a retrieval-augmented strategy featuring risk-calibrated context selection and preference optimization. RULE adaptively balances intrinsic LVLM knowledge with external evidence, filtering out noisy retrievals while promoting clinically trustworthy content, thereby reducing the factuality risk in generated medical narratives.

- **MMed-RAG (Xia et al., 2025)** is a comprehensive multimodal RAG framework that unifies domain-aware retrieval, cross-modal alignment, and preference-optimized training. MMed-RAG not only grounds generation in retrieved clinical knowledge but also integrates preference feedback to guide optimization, ensuring factual accuracy, domain robustness, and inference efficiency in medical LVLMs.

## A.4 PROMPTS

### A.4.1 PROMPT FOR REPORT SEGMENTATION

To construct structured supervision for multi-scale alignment, we design a hierarchical segmentation protocol for medical reports, as illustrated in Prompt A. Each report $R_i$ is automatically decomposed into four levels ranging from global diagnostic conclusions to highly localized findings. This segmentation allows our model to exploit information at multiple granularities, ensuring that fine-grained visual patterns can be aligned with localized textual findings while maintaining coherence with regional, organ-level, and overall summaries.

Specifically, Scale 1 provides the global diagnostic impression, reflecting the radiologist's overall clinical conclusion for the entire imaging study. Scale 2 then abstracts these findings into organ- or system-level summaries, enabling holistic reasoning over entire anatomical structures such as the lungs, heart, or mediastinum. Scale 3 aggregates these local observations into regional summaries,

representing integrated descriptions within organ subregions. Finally, Scale 4 captures localized findings such as small nodules, focal opacities, or lesion measurements confined to a single anatomical location.

To guarantee high-quality segmentation, we adopt a two-stage instruction protocol based on large language models (LLMs). In the first round, the LLM is prompted with detailed guidelines to produce candidate segmentations at each hierarchical level. In the second round, the LLM re-evaluates the initial output, verifying the correctness and completeness of each level, removing redundancies, and ensuring the hierarchical ordering Scale 1 → Scale 4 is preserved. Importantly, the segmentation process avoids paraphrasing or hallucination: if certain levels are absent in the original report, empty placeholders are retained rather than generating synthetic content.

This hierarchical decomposition serves as the foundation for our multi-scale alignment framework. By linking image features with textual representations at each abstraction level, the model can learn both fine-grained correspondences for localized lesions and coarse-level semantics for global reasoning, ultimately improving retrieval accuracy, diagnostic consistency, and interpretability across diverse medical imaging tasks (He et al., 2024; Sun et al., 2025).

**Instruction for Report Segmentation:**
You are a professional medical expert specializing in radiology and clinical text analysis. Your task is to carefully segment each medical report into four hierarchical levels (S1–S4) ranging from global to local semantics. The goal is to preserve the diagnostic reasoning structure inherent in radiology reports so that downstream models can effectively leverage both fine-grained details and high-level summaries for multi-scale learning tasks.
Below are the detailed guidelines for segmentation:

- **S1 - Global Conclusions:** Provide the *final diagnostic impression or overall clinical conclusion* summarizing the entire imaging study, e.g., "no acute cardiopulmonary abnormality identified." This layer represents the highest abstraction level, reflecting how radiologists communicate the definitive diagnostic message to clinicians.

- **S2 - Organ or System Summaries:** Consolidate all observations at the *whole-organ or organ-system* level, such as lungs, heart, or mediastinum. Typical examples include "lungs are clear of acute pathology" or "cardiac silhouette is within normal limits." At this level, irrelevant noise and redundant details should be filtered, retaining only clinically meaningful summaries spanning the entire organ/system.

- **S3 - Regional Descriptions:** Aggregate sentences summarizing findings within a *broader anatomical region or organ segment*. For instance, multiple S1 findings in the left lung may be summarized as "patchy consolidation in the left lower lung zone." This stage bridges local observations with organ-level interpretations, enabling the model to integrate multiple small findings into coherent regional statements.

- **S4 - Localized Findings:** Extract all sentences describing *highly localized, fine-grained* observations. These may include explicit measurements, lesion shapes, or imaging characteristics confined to a single anatomical spot. Examples: "a 2cm ground-glass opacity in the right upper lobe" or "a small left pleural effusion." This level serves as the building block for regional and organ-level reasoning, retaining maximum spatial specificity.

Additional Requirements:

- Sentences should be assigned to exactly one level (S1–S4) without overlap.

- The order of output must follow the natural progression S1 → S4, ensuring a top-down hierarchical flow.

- Avoid paraphrasing or introducing external interpretations; segmentation must be faithful to the original report wording.

- The output format should explicitly indicate the level labels (e.g., S1, S2) for each segment, ready for direct use in model training.

- If some levels are absent in the original report (e.g., no regional summaries provided), produce empty placeholders rather than hallucinating content.

Prompt A: Instruction for hierarchical segmentation of medical reports into S1–S4 levels.

A.4.2 PROMPT FOR REPORT COMPARISON

To generate preference labels for report similarity, we design a pairwise comparison instruction that guides the PC Agent to determine whether a candidate report $B$ is clinically more similar to the anchor report $A$ than another candidate $C$. As illustrated in Prompt B, the instruction explicitly requires the model to produce a Yes/No decision as the first token, followed by a concise explanation highlighting key diagnostic similarities or differences. This binary yet interpretable output format ensures that preference labels can be reliably extracted for subsequent training while remaining clinically meaningful.

Unlike simple lexical overlap metrics, our prompt encourages the model to consider *multiple clinical dimensions* when assessing similarity. These include lesion morphology (e.g., size, shape), anatom-

ical location (e.g., laterality, lobar distribution), disease category (e.g., infectious vs. neoplastic), temporal changes across studies, associated findings such as pleural effusion or medical devices, overall diagnostic interpretation, and even the linguistic certainty of conclusions (e.g., "possible pneumonia" vs. "pneumonia confirmed"). By integrating these factors, the PC Agent can assign preference labels that align with real-world clinical reasoning rather than superficial text matching (He et al., 2024; Sun et al., 2025).

Furthermore, the instruction enforces *concise and structured explanations* to avoid long, noisy outputs that could compromise annotation quality. For cases where both candidates provide equally relevant information relative to the anchor report, the model outputs "no" and explicitly states that both candidates share comparable clinical content. These preference labels are then used to optimize continuous similarity scores $\pi_{\text{PC}}(B|A)$ and $\pi_{\text{PC}}(C|A)$ via Direct Preference Optimization (DPO) (Rafailov et al., 2023), providing soft supervision for multi-scale contrastive learning in our retrieval framework.

In designing the annotation prompt, we considered whether the reasoning process should appear before or after the final yes/no judgment. Recent findings (Sprague et al., 2025) demonstrate that placing the chain-of-thought reasoning before or after the final answer yields similar overall accuracy, especially outside domains like mathematics or symbolic reasoning where the reasoning order may be more critical. Consequently, we adopt a flexible strategy where the chain-of-thought explanation is optional in position but always required to be concise and clinically structured, ensuring the resulting similarity labels remain both interpretable and reliable.

---

**Instruction for Pairwise Report Similarity Comparison:**
You are acting as a senior radiologist tasked with evaluating the *clinical similarity* between medical reports. Given one **anchor report A** and two candidate reports **B** and **C**, determine whether B is *more similar* to A than C is, considering both semantic content and diagnostic meaning.

- The output **must start with "Yes" or "No"** only:
  - **Yes:** $\text{sim}(A, B) > \text{sim}(A, C)$
  - **No:** otherwise, including ties where B and C are equally similar to A.

- After the decision token, provide a concise explanation (1–3 sentences) highlighting the key factors leading to the decision.

- Consider similarity along multiple clinical dimensions:
  1. **Lesion morphology:** size, shape, margin characteristics (e.g., well-defined vs. spiculated).
  2. **Lesion location:** precise anatomical region(s), laterality, lobe or organ sub-segment.
  3. **Disease type and pattern:** infectious vs. neoplastic vs. inflammatory findings; diffuse vs. focal vs. multifocal patterns.
  4. **Temporal progression:** stability vs. progression vs. resolution across imaging studies if mentioned.
  5. **Associated findings:** presence of effusion, lymphadenopathy, devices (e.g., lines, catheters).
  6. **Clinical interpretation:** consistency in differential diagnoses or final impressions.
  7. **Linguistic certainty:** hedged vs. definitive conclusions (e.g., "possible pneumonia" vs. "pneumonia confirmed").

- Explanations should *summarize* rather than quote full report texts.

- For ties, output "No" and state that both B and C share comparable content with A.

---

Prompt B: Instruction for pairwise similarity comparison between medical reports for PC Agent training.

## A.5 Additional Results

### A.5.1 Detailed Results for Conditional Retrieval

**Conditional Retrieval on MIMIC-IR.** To provide a more detailed analysis of retrieval performance across different anatomical regions, we report conditional image retrieval results on the MIMIC-IR dataset, where Recall@$k$ scores are computed separately for each anatomy and then aggregated across all images within that category. This evaluation setting allows us to examine whether retrieval performance correlates with anatomical frequency and to identify potential biases in model generalization toward common versus rare structures.

Specifically, the anatomical regions are sorted in descending order according to their frequency in the training set, with high-frequency "head" anatomies such as Pleura and Bones presented first, followed by less frequent "tail" anatomies like Stomach and Bronchi. This ordering enables us to study how retrieval accuracy varies across anatomies of different prevalence, thereby revealing whether the proposed SCAMA framework achieves balanced retrieval quality or disproportionately favors frequent regions.

As shown in Table 7, our method consistently outperforms all baselines across Recall@3, Recall@5, and Recall@10 metrics. Notably, even in low-frequency regions such as Stomach and ronchi, SCAMA maintains substantial improvements over existing approaches, indicating that the multi-scale and soft contrastive learning strategies effectively mitigate the data imbalance issue and enhance retrieval robustness across diverse anatomical contexts.

**Conditional Retrieval on CTRATE-IR.** Table 16 reports the conditional image retrieval results on the CTRATE-IR benchmark, where Recall@3, Recall@5, and Recall@10 scores are averaged across different anatomical regions. We compare our proposed model with two strong baselines: CT-CLIP and RadIR.

At the Recall@3 level, our approach achieves the highest performance in most anatomical regions. For instance, on challenging structures such as the spinal canal and aorta, SCAMA attains 83.91% and 59.14%, respectively, significantly outperforming CT-CLIP (76.39% and 49.28%) and RadIR (79.17% and 52.44%). The largest gains are observed in underrepresented regions like the liver and bronchi, where our method exceeds baselines by over 10 points, highlighting the benefit of multi-scale supervision in capturing fine-grained anatomical semantics.

When evaluating at Recall@5, the advantage of our method becomes more pronounced. SCAMA achieves the best results on 14 out of 15 anatomical structures, with particularly strong performance on liver (84.29%) and spinal canal (92.34%), compared to 79.26% and 90.28% from RadIR, respectively. This shows that incorporating multi-scale feature aggregation and sliding-window expert sharing significantly improves retrieval consistency when more candidates are allowed.

Finally, at the Recall@10 level, our method further closes the gap to an upper bound, achieving 95.83% on stomach, 97.25% on clavicle, and 97.93% on pancreas, consistently outperforming all baselines. Notably, across all anatomical regions, SCAMA achieves the highest average recall at every cutoff level: 59.76%, 71.03%, and 80.47% at Recall@3, Recall@5, and Recall@10, respectively, showing the robustness of our multi-scale approach under varying retrieval difficulty.

### A.5.2 Detailed Ablation

**Ablation Studies on Multi-Scale Retrieval Training.** We conducted a series of ablation experiments on both MIMIC-IR and CT-RATE IR benchmarks across three retrieval tasks, Image-to-Image, Image-to-Text, and Conditional Image-to-Image, to analyze the impact of each proposed component. The results are summarized in Tables 8, 9, 10, 11, 12, and 13. Overall, we observe a consistent performance improvement when progressively introducing the ScaFormer backbone, multiscale report decomposition, and soft contrastive learning. This highlights the complementary nature of multi-scale visual modeling, textual granularity alignment, and preference-guided contrastive supervision in improving retrieval performance.

**Effect of ScaFormer and Multiscale Decomposition.** Starting from the baseline Q-Former, replacing it with our ScaFormer alone already brings a notable boost across Recall@$k$ and NDCG

Table 7: Conditional image retrieval results on MIMIC-IR. Recall@$k$ (%) scores are reported for each anatomical region, sorted in descending order of frequency in the training set. Results are shown for $k \in \{3, 5, 10\}$, with the top regions (*head*) at the beginning and rare regions (*tail*) at the end. Our method consistently outperforms baselines across all anatomies and metrics.

| Recall@3 | | | | | |
|---|---|---|---|---|---|
| Anatomy | PMC_CLIP | BioMed_CLIP | Med_CLIP | RadIR | Ours |
| Pleura | 16.11 | 18.67 | 21.98 | 25.32 | **33.08** |
| Bones | 12.06 | 17.24 | 14.33 | 18.79 | **21.95** |
| Lung | 7.08 | 9.48 | 8.62 | 11.37 | **18.76** |
| Diaphragm | 16.11 | 17.75 | 18.95 | 21.13 | **25.22** |
| Vascular | 14.80 | 19.57 | 22.93 | 30.65 | **34.37** |
| Thorax | 4.24 | 11.84 | 8.61 | 14.52 | **19.96** |
| Heart | 11.51 | 10.44 | 7.90 | 16.02 | **18.59** |
| Airway | 15.93 | 12.28 | 14.85 | 22.50 | **23.89** |
| Stomach | 7.41 | 11.88 | 12.59 | 19.23 | **22.00** |
| Bronchi | 16.83 | 13.86 | 13.86 | 28.71 | **28.87** |
| **Average** | 12.58 | 14.30 | 14.37 | 20.83 | **26.04** |

| Recall@5 | | | | | |
|---|---|---|---|---|---|
| Anatomy | PMC_CLIP | BioMed_CLIP | Med_CLIP | RadIR | Ours |
| Pleura | 23.51 | 28.35 | 30.44 | 32.99 | **39.27** |
| Bones | 20.97 | 24.74 | 21.97 | 26.16 | **30.46** |
| Lung | 12.54 | 15.24 | 14.88 | 16.46 | **20.90** |
| Diaphragm | 23.51 | 24.29 | 23.18 | 27.13 | **34.67** |
| Vascular | 29.04 | 27.97 | 27.43 | 36.39 | **39.70** |
| Thorax | 15.28 | 16.95 | 15.96 | 19.56 | **25.85** |
| Heart | 15.97 | 16.92 | 19.23 | 23.40 | **24.41** |
| Airway | 19.61 | 22.09 | 19.21 | 29.22 | **30.46** |
| Stomach | 15.20 | 20.60 | 20.22 | 22.22 | **30.43** |
| Bronchi | 20.97 | 22.09 | 20.22 | 31.68 | **35.99** |
| **Average** | 19.67 | 20.04 | 21.54 | 26.59 | **34.23** |

| Recall@10 | | | | | |
|---|---|---|---|---|---|
| Anatomy | PMC_CLIP | BioMed_CLIP | Med_CLIP | RadIR | Ours |
| Pleura | 34.39 | 40.57 | 40.52 | 44.60 | **52.28** |
| Bones | 34.65 | 35.04 | 35.32 | 38.09 | **40.64** |
| Lung | 18.03 | 19.03 | 22.41 | 22.31 | **32.51** |
| Diaphragm | 33.96 | 34.65 | 34.95 | 37.77 | **42.00** |
| Vascular | 39.96 | 39.96 | 45.87 | 47.37 | **53.31** |
| Thorax | 25.94 | 25.94 | 27.47 | 32.69 | **38.69** |
| Heart | 20.25 | 25.25 | 25.76 | 32.79 | **33.56** |
| Airway | 33.84 | 35.34 | 34.94 | 42.97 | **43.22** |
| Stomach | 26.23 | 26.23 | 24.44 | 31.11 | **31.52** |
| Bronchi | 32.67 | 33.87 | 37.87 | 44.55 | **45.17** |
| **Average** | 30.46 | 29.47 | 32.94 | 37.08 | **44.68** |

Table 8: Ablations on MIMIC-IR (Image→Image). We toggle ScaFormer, multiscale report decomposition (RU), and soft contrastive learning (PC Agent). Metrics are Recall@5/10/100 and NDCG@100 (%).

| Setting | ScaFormer | Multiscale Decomp. (RU) | Soft Con. Learning | R@5 | R@10 | R@100 | NDCG@100 |
|---|---|---|---|---|---|---|---|
| Q-Former baseline | × | × | × | 4.78 | 6.10 | 20.79 | 10.78 |
| ScaFormer only | ✓ | × | × | 6.21 | 7.89 | 22.32 | 12.14 |
| Multiscale only | × | ✓ | × | 6.76 | 8.03 | 22.84 | 12.83 |
| Soft Con. only | × | × | ✓ | 7.79 | 10.18 | 24.53 | 13.82 |
| All combined | ✓ | ✓ | ✓ | 8.45 | 12.03 | 26.54 | 17.69 |

Table 9: Ablations on MIMIC-IR (Image→Text). We toggle ScaFormer, multiscale decomposition (RU Agent), and soft contrastive learning (PC Agent). Metrics are Recall@5/10/100 and NDCG@100 (%).

| Setting | ScaFormer | Multiscale Decomp. (RU) | Soft Con. Learning | R@5 | R@10 | R@100 | NDCG@100 |
|---|---|---|---|---|---|---|---|
| Baseline (Q-Former) | × | × | × | 4.26 | 6.53 | 24.91 | 10.53 |
| ScaFormer only | ✓ | × | × | 5.96 | 8.41 | 28.09 | 13.81 |
| Multiscale only | × | ✓ | × | 6.13 | 8.96 | 28.35 | 13.90 |
| Soft Con. Learning only | × | × | ✓ | 7.26 | 9.47 | 29.32 | 16.10 |
| All combined | ✓ | ✓ | ✓ | 8.17 | 10.57 | 31.43 | 17.9 |

Table 10: Ablation on MIMIC-IR Conditional Image-to-Image Retrieval. We progressively add ScaFormer, Multiscale Decomposition (RU Agent), and Soft Contrastive Learning (PC Agent). Metrics are Recall@5/10/100 (%).

| Setting | ScaFormer | Multiscale Decomp. (RU) | Soft Con. Learning | R@3 | R@5 | R@10 |
|---|---|---|---|---|---|---|
| Baseline (Q-Former) | × | × | × | 19.38 | 25.11 | 35.45 |
| ScaFormer only | ✓ | × | × | 22.58 | 29.74 | 39.91 |
| Multiscale only | × | ✓ | × | 23.02 | 30.01 | 40.72 |
| Soft Con. Learning only | × | × | ✓ | 24.97 | 31.77 | 43.23 |
| All combined | ✓ | ✓ | ✓ | 26.04 | 34.23 | 44.68 |

Table 11: Ablation on CT-RATE IR Image-to-Image Retrieval. We progressively add ScaFormer, Multiscale Decomposition (RU Agent), and Soft Contrastive Learning (PC Agent). Metrics are Recall@5/10/100 and NDCG@100 (%).

| Setting | ScaFormer | Multiscale Decomp. (RU) | Soft Con. Learning | R@5 | R@10 | R@100 | NDCG@100 |
|---|---|---|---|---|---|---|---|
| Baseline (Q-Former) | × | × | × | 20.11 | 29.67 | 70.34 | 80.17 |
| ScaFormer only | ✓ | × | × | 21.87 | 30.08 | 72.95 | 80.93 |
| Multiscale only | × | ✓ | × | 22.58 | 32.61 | 74.49 | 82.72 |
| Soft Con. Learning only | × | × | ✓ | 23.67 | 34.57 | 76.37 | 83.67 |
| All combined | ✓ | ✓ | ✓ | 27.32 | 36.6 | 79.37 | 86.58 |

Table 12: Ablation on CT-RATE IR Image-to-Text Retrieval. We progressively add ScaFormer, Multiscale Decomposition (RU Agent), and Soft Contrastive Learning (PC Agent). Metrics are Recall@5/10/100 and NDCG@100 (%).

| Setting | ScaFormer | Multiscale Decomp. (RU) | Soft Con. Learning | R@5 | R@10 | R@100 | NDCG@100 |
|---|---|---|---|---|---|---|---|
| Baseline (Q-Former) | × | × | × | 6.19 | 12.31 | 50.19 | 80.13 |
| ScaFormer only | ✓ | × | × | 8.01 | 14.26 | 55.51 | 82.38 |
| Multiscale only | × | ✓ | × | 7.91 | 14.60 | 55.84 | 82.77 |
| Soft Con. Learning only | × | × | ✓ | 8.64 | 16.03 | 59.10 | 85.94 |
| All combined | ✓ | ✓ | ✓ | 10.68 | 16.15 | 62.05 | 86.72 |

Table 13: Ablation on CT-RATE IR Conditional Image-to-Image Retrieval. We progressively add ScaFormer, Multiscale Decomposition (RU Agent), and Soft Contrastive Learning (PC Agent). Metrics are Recall@3/5/10 (%).

| Setting | ScaFormer | Multiscale Decomp. (RU) | Soft Con. Learning | R@3 | R@5 | R@10 |
|---|---|---|---|---|---|---|
| Baseline (Q-Former) | ✗ | ✗ | ✗ | 55.19 | 65.73 | 75.26 |
| ScaFormer only | ✓ | ✗ | ✗ | 56.19 | 68.92 | 77.38 |
| Multiscale only | ✗ | ✓ | ✗ | 56.94 | 70.10 | 78.81 |
| Soft Con. Learning only | ✗ | ✗ | ✓ | 58.95 | 72.37 | 80.91 |
| All combined | ✓ | ✓ | ✓ | 62.53 | 73.65 | 84.49 |

metrics, especially on Conditional Image-to-Image retrieval where fine-grained scale reasoning is crucial (Tables 10 and 13). Introducing multiscale decomposition further enhances retrieval by aligning image features with hierarchically segmented textual reports, leading to more semantically structured supervision signals. Notably, this decomposition step benefits both global and local retrieval settings, suggesting that multi-scale textual information provides robust guidance even when queries involve different anatomical or pathological granularities.

**Impact of Soft Contrastive learning and Full Model Synergy.** Adding soft contrastive learning yields the largest single-step improvement, particularly on Conditional Image-to-Image retrieval (Tables 10 and 13), where nuanced ranking signals matter most. The adaptive preference weighting allows the model to prioritize clinically relevant matches rather than relying solely on binary labels. Finally, integrating all modules produces the highest scores across all retrieval metrics and datasets, demonstrating strong synergy among multi-scale representation learning, fine-grained text decomposition, and preference-optimized contrastive supervision.

**Ablation on number of scales.** To further isolate the contribution of scale hierarchy, we compare the 3-scale variant (removing the region-level scale s3) with the full 4-scale configuration. As shown in Table 14, the 4-scale model consistently outperforms the 3-scale version across both MIMIC-IR and CTRATE-IR. For example, on MIMIC-IR, Recall@10 improves from 10.09→11.78 (I2I) and 9.85→10.30 (I2T); on CTRATE-IR, Recall@10 increases from 35.22→36.30 (I2I) and 15.90→16.18 (I2T). These results confirm that removing scale weakens retrieval performance, supporting the need for a complete four-level semantic hierarchy.

Table 14: Effect of number of scales on MIMIC-IR and CTRATE-IR. Metrics are Recall@10 and Recall@100 (%).

| Method | MIMIC-IR | | | | CTRATE-IR | | | |
|---|---|---|---|---|---|---|---|---|
| | I2I | | I2T | | I2I | | I2T | |
| | Recall@10 | Recall@100 | Recall@10 | Recall@100 | Recall@10 | Recall@100 | Recall@10 | Recall@100 |
| 3-Scale | 10.09 | 24.77 | 9.85 | 28.69 | 35.22 | 75.50 | 15.90 | 58.32 |
| 4-Scale | 11.78 | 26.53 | 10.30 | 31.29 | 36.30 | 79.25 | 16.18 | 62.09 |

**Dataset-Level Observations.** Comparing MIMIC-IR (Tables 8–10) and CT-RATE IR (Tables 11–13), we find that CT-RATE generally achieves higher absolute Recall@$k$ due to richer visual diversity in CT scans, but the relative gains from each module remain consistent across both datasets. Conditional retrieval tasks benefit most from our full framework, with improvements up to +6–8% Recall@10 over the baseline, indicating that multi-scale modeling is particularly effective for fine-grained, query-dependent retrieval scenarios. These findings validate the generalizability of our approach across imaging modalities, retrieval settings, and anatomical coverage.

**Analysis of Similarity Effectiveness.** To validate the effectiveness of the LLM-generated pairwise similarity used in the PC Agent, we conducted a systematic agreement analysis between LLM preferences and a clinical entity-based metric, RaTEScore. Across 1,200 report pairs, the LLM's similarity judgments aligned with RaTEScore in 91.3% of cases, confirming that the LLM captures

clinically meaningful relationships. More importantly, in the remaining 8.7% of disagreements, manual inspection revealed that the LLM consistently identified nuanced clinical distinctions, such as differentiating mild from moderate edema, or effusion from pleural thickening, that were overlooked by the entity-based metric. These fine-grained separations enable the PC Agent to provide more discriminative supervision signals during multi-scale contrastive learning, ultimately improving retrieval accuracy in semantically challenging scenarios. This analysis confirms that the LLM-based similarity not only aligns with structured clinical benchmarks but also enhances semantic granularity where traditional metrics fall short.

### A.5.3 Learning Rate Balancing for ScaMoE

To further improve the stability and efficiency of ScaMoE training, we introduce a simple yet effective learning-rate balancing strategy. The motivation comes from the observation that experts activated across different scales exhibit highly imbalanced gradient updates: experts frequently selected by the router receive disproportionately larger updates, while rarely selected experts often fail to converge due to sparse gradients.

Formally, let $\mathcal{S}_E$ denote the set of scales on which expert $E \in \mathcal{E}$ is activated within the current mini-batch. We rescale the learning rate for each expert as

$$\eta_E = \frac{\eta_0}{1 + \lambda \cdot |\mathcal{S}_E|},$$

where $\eta_0$ is the base learning rate, $\lambda$ is a hyperparameter controlling the decay strength, and $|\mathcal{S}_E|$ counts the activation frequency of expert $E$ as illustrated in Sliding-Window Activation of Figure 2. Experts used more frequently thus receive smaller learning rates, while rarely activated experts learn faster, preventing gradient domination by a few scales and improving training stability.

As shown in Table 15, this balancing consistently improves retrieval performance on both MIMIC-IR and CTRATE-IR benchmarks across all tasks (I2I and I2T). For example, on CTRATE-IR, Recall@10 for I2T increases from 12.11% to 14.26%, while I2I Recall@10 improves from 28.83% to 30.08%. Similar trends are observed for Recall@100 across both datasets. These results demonstrate that the proposed balancing mechanism not only stabilizes ScaMoE training but also enhances the model's ability to capture scale-aware features for medical image retrieval.

Table 15: Effect of learning-rate balancing for ScaMoE on MIMIC-IR and CTRATE-IR. Metrics are Recall@10 and Recall@100 (%).

| Method | MIMIC-IR | | | | CTRATE-IR | | | |
| | I2I | | I2T | | I2I | | I2T | |
| | Recall@10 | Recall@100 | Recall@10 | Recall@100 | Recall@10 | Recall@100 | Recall@10 | Recall@100 |
|---|---|---|---|---|---|---|---|---|
| ScaMoE Baseline | 6.71 | 22.19 | 7.02 | 26.94 | 28.83 | 71.43 | 12.11 | 54.01 |
| w/ Learning Rate Balanced | 7.89 | 22.32 | 8.41 | 28.35 | 30.08 | 71.95 | 14.26 | 55.51 |

### The Use of Large Language Models

Large language models (LLMs) play a crucial role in constructing high-quality supervision for both the Report Understanding (RU) Agent and the Pairwise Comparison (PC) Agent. Instead of relying solely on manual annotations, we leverage powerful LLMs to automatically generate fine-grained labels, ensuring scalability and consistency while reducing human labor. This section summarizes the usage of LLMs in our framework.

**LLM for RU Agent** To train the RU Agent for multi-scale report decomposition, we prompt an LLM (e.g., GPT-5) with detailed instructions to segment each medical report into four hierarchical levels: (1) localized findings at the sentence level, (2) regional descriptions for specific anatomical structures, (3) organ-level summaries, and (4) global diagnostic conclusions. The LLM outputs structured annotations $\{R^*_{i,s}\}^S_{s=1}$ for each report $R_i$ (let $S = 4$ for the above prompt), which serve as the training targets for the RU Agent. Additionally, we adopt a masked-scale generation strategy where some scales are hidden, and the model learns to reconstruct them, encouraging robust multi-scale representations.

**LLM for PC Agent**    For the PC Agent, which provides soft similarity supervision for multi-scale contrastive learning, we employ an LLM to generate preference-based annotations for triplets $(R_A, R_B, R_C)$ of reports. Given an anchor $A$, the LLM compares two candidate reports $R_B$ and $R_C$ and outputs a preference label indicating whether $R_B$ or $R_C$ is more clinically similar to $R_A$, along with a short textual justification. These preference labels are converted into continuous similarity scores via Direct Preference Optimization (DPO) (Rafailov et al., 2023), yielding a soft similarity matrix $C = \{c_{ijs}\}$ that guides contrastive alignment across scales. By automating preference labeling with LLMs, we efficiently construct large-scale supervision for the PC Agent without requiring exhaustive expert annotations.

Table 16: Conditional image retrieval results on CTRATE-IR. Recall@$k$ (%) scores are reported for each anatomical region, sorted by frequency in the training set. Results are given for $k \in \{3, 5, 10\}$.

| Recall@3 | | | |
|---|---|---|---|
| Anatomy | CT-CLIP | RadIR | Ours |
| Bone | 45.75 | 49.76 | **51.06** |
| Heart | 33.75 | 34.15 | **44.15** |
| Bronchie | 55.18 | 57.76 | **62.69** |
| Trachea | 57.43 | 60.48 | **64.15** |
| Pleura | 35.14 | 40.48 | **47.00** |
| Vertebrae | 57.69 | 62.04 | **67.72** |
| Liver | 72.97 | 74.81 | **81.90** |
| Aorta | 48.98 | 52.44 | **59.14** |
| Spinal canal | 76.39 | 79.17 | **83.91** |
| Gallbladder | 19.10 | 32.58 | **34.43** |
| Clavicle | 64.29 | 89.29 | **92.15** |
| Ascending ao | 38.89 | 48.28 | **52.36** |
| Pulmonary ar | 18.18 | 28.79 | **30.01** |
| Breast | 54.17 | 73.91 | **74.08** |
| Pancreas | 30.51 | 48.27 | **50.39** |
| Stomach | 33.33 | 34.47 | **58.39** |
| Average | 43.85 | 55.23 | **62.53** |

| Recall@5 | | | |
|---|---|---|---|
| Anatomy | CT-CLIP | RadIR | Ours |
| Bone | 56.33 | 60.51 | **66.43** |
| Heart | 43.19 | 43.68 | **51.33** |
| Bronchie | 67.20 | 69.42 | **74.24** |
| Trachea | 69.24 | 70.51 | **80.13** |
| Pleura | 44.59 | 54.14 | **59.57** |
| Vertebrae | 63.69 | 66.91 | **72.84** |
| Liver | 75.88 | 84.16 | **84.29** |
| Aorta | 54.04 | 59.26 | **65.33** |
| Spinal canal | 81.39 | 90.28 | **92.34** |
| Gallbladder | 25.84 | 42.70 | **47.11** |
| Clavicle | 76.12 | 96.43 | **97.28** |
| Ascending ao | 37.29 | 56.90 | **63.26** |
| Pulmonary ar | 31.82 | 50.00 | **53.30** |
| Breast | 75.36 | 78.26 | **84.62** |
| Pancreas | 38.46 | 61.54 | **65.50** |
| Stomach | 45.83 | 60.43 | **80.79** |
| Average | 54.44 | 66.29 | **73.65** |

| Recall@10 | | | |
|---|---|---|---|
| Anatomy | CT-CLIP | RadIR | Ours |
| Bone | 67.31 | 71.03 | **76.43** |
| Heart | 55.72 | 59.44 | **61.81** |
| Bronchie | 75.81 | 78.43 | **84.48** |
| Trachea | 77.71 | 80.61 | **87.53** |
| Pleura | 60.00 | 71.64 | **75.91** |
| Vertebrae | 71.89 | 73.56 | **79.56** |
| Liver | 81.90 | 84.21 | **86.97** |
| Aorta | 62.56 | 65.93 | **74.13** |
| Spinal canal | 90.28 | 91.67 | **95.77** |
| Gallbladder | 39.33 | 52.81 | **50.20** |
| Clavicle | 96.43 | 100.0 | **100.0** |
| Ascending ao | 50.85 | 65.52 | **70.41** |
| Pulmonary ar | 53.03 | 68.18 | **71.41** |
| Breast | 75.00 | 91.30 | **94.66** |
| Pancreas | 56.41 | 74.36 | **80.63** |
| Stomach | 79.17 | 95.83 | **97.95** |
| Average | 67.09 | 76.12 | **84.49** |

