# OpenReview forum: "MedSCAMA: Medical SCale-Aware Multi-Agent Framework for Medical Image Retrieval and Retrieval-Augmented Generation"
_ICLR.cc/2026/Conference — ICLR 2026 Conference Desk Rejected Submission_

### Official Review · Reviewer_fL5n · 2025-10-26

**Soundness:** 3
**Presentation:** 1
**Contribution:** 3
**Rating:** 4
**Confidence:** 3

**Summary:**

The paper proposes MedSCAMA, a scale-aware multi-agent framework for medical image retrieval and retrieval-augmented generation. MedSCAMA introduces a ScaFormer for multi-scale visual encoding with a sliding-window Mixture-of-Experts, a Report Understanding (RU) Agent for hierarchical report decomposition, a Pairwise Comparison (PC) Agent trained with soft contrastive loss and Direct Preference Optimization, and a Question Answering (QA) Agent for adaptive evidence retrieval and reasoning. Experiments on medical retrieval and QA benchmarks demonstrate improved performance over existing baselines.

**Strengths:**

1.  The idea of ScaFormer is sound. Introducing a scale-aware Mixture-of-Experts design would enable multi-resolution feature extraction, a crucial aspect in medical imaging.

2. Experimental results on medical retrieval and QA tasks show consistent improvements over strong baseline methods.

**Weaknesses:**

1. Over-complex designs. Given the complexity of model designs, it is suggested to improve the writing for better readability.

2. Figure quality. It is suggested to enhance the quality of Figure 2 for better clarity.

Overall, the proposed framework seems effective. However, it is strongly suggested to improve the writing for better readability.

**Questions:**

please see weaknesses.

---

> ### Author Response · Authors · 2025-11-21
> **Response to Reviewer fL5n**
>
> Dear Reviewer fL5n,
>
> Thank you for the summary comments. We are fully committed to improving readability and presentation quality, and we have updated our manuscript based on other reviewers' comments. We would be grateful if the reviewer could review again and indicate specific segments that were unclear, so that our revision can be maximally useful.

---

> > ### Comment · Reviewer_fL5n · 2025-11-22
> > **Thanks for the reply**
> >
> > Thank the authors for the reply. I appreciate your great efforts during the rebuttal to improve the readability. However, please allow me to highlight the following points that support my current negative assessment:
> >
> > - **Limited novelty.** As also mentioned by Reviewer J5rp, this work seems to be a complex combination of current methods, including MoE, DPO, and contrastive learning. While these engineering efforts indeed lead to better performance, they lack rigorous theoretical analysis and insights.
> >
> > - **Figure quality.** It is again suggested to improve the quality of Figure 2 (e.g., colors and layout). Prior ICLR papers may serve as useful references.
> >
> > - **Fragility of the system design.** Due to the complex nature of the system, there are many hyperparameters that might make it ineffective for real-world deployment. For example, directly using a teacher model (GPT5 here) for medical report segmentation (quantitative human expert judgement is necessary), the number of scales, and the factor to balance multi-scale representation learning and DPO.
> >
> > Again, I appreciate the engineering efforts made in this paper. However, the overall scientific contribution seems somewhat limited for ICLR. Given the above reasons, I would like to maintain my assessment.

---

> > > ### Author Response · Authors · 2025-11-27
> > > **Follow-up Response to Reviewer fL5n (Part 1/2)**
> > >
> > > We thank the reviewer for the follow-up. We respectfully disagree with the assessment regarding "limited novelty" and "fragility". We believe these concerns overlook the specific domain challenges our architecture addresses and the quantitative evidence provided. We clarify these points below.
> > >
> > > ---
> > >
> > > > **W1.** Limited novelty.
> > >
> > > We argue that our novelty lies not in simply combining existing methods but in addressing fundamental limitations via specific architecture designs. To articulate our novelty from a different perspective, our primary goal is to resolve the mismatch in multi-scale vision and language representations for medical vision-language models. We addressed this via two synergistic pathways:
> > >
> > > **Pathway 1**: Better multi-scale feature **extraction**. We introduced two novel components for this purpose:
> > >
> > > 1. ScaFormer (recognized by *Reviewer fL5n*). It consists of multi-scale feature extraction, scale-wise cross-attention, and a scale-aware MoE, for multi-scale visual feature extraction.
> > > 2. RU Agent. It is designed to distill the text segmentation capabilities of LLM, converting unstructured reports to multi-scale text features.
> > >
> > > **Pathway 2**: Better multi-scale feature **alignment**. We also proposed two novel components for this purpose:
> > > 1. PC Agent (recognized by *Reviewer 6n5C* and *Reviewer J5rp*). It works by combining soft contrastive learning and active data selection with DPO to align visual and textual features.
> > > 2. QA Agent. It integrates the aligned features into the RAG pipeline, enabling adaptive retrieval and evidence-grounded generation.
> > >
> > > While the MoE, DPO, and contrastive learning were the initial building blocks of MedSCAMA, our specialized improvements constitute a distinct aspect of the novelty.
> > >  - We redesigned the MoE with sliding-window operations to ensure that each expert could specialize in a distinct combination of scales.
> > >  - We adapted contrastive learning in a soft-label setting, and it is working with DPO in an end-to-end multi-modality training process to align vision and language features.
> > >
> > > In summary, as also discussed in the paper and in our reply to *Reviewer J5rp*, our novelty lies not just in those components, but also in the orchestration among agents (and ScaFormer). In our extensive ablation studies, we have provided evidence that these components and the system are effective across multiple datasets and several tasks (image retrieval, VQA, and report generation), achieving state-of-the-art performance. Our contribution has also been generally recognized by *Reviewer 6n5C*.
> > >
> > > ---
> > >
> > > > **W2.** Figure quality.
> > >
> > > Thank you for your continuous suggestions on the colors and layout of Figure 2. We initially referred to several ICLR papers for the design, including MMed-RAG [Ref-1], Speculative RAG [Ref-2], and V-RAG [Ref-3], but we acknowledge that the visual delivery could be further enhanced. We have make our best effort to modify the whole pipeline Figure 2, refining both the color scheme and layout structure as requested by the reviewer. We kindly invite you to check the revised version. Please let us know if there is anything we could improve further regarding this figure.
> > >
> > > [Ref-1] Xia, Peng, et al. "MMed-RAG: Versatile Multimodal RAG System for Medical Vision Language Models." The Thirteenth International Conference on Learning Representations (ICLR). 2025.
> > >
> > > [Ref-2] Wang, Zilong, et al. "Speculative RAG: Enhancing Retrieval Augmented Generation through Drafting." The Thirteenth International Conference on Learning Representations.
> > >
> > > [Ref-3] Chen, Jun, et al. "Document haystacks: Vision-language reasoning over piles of 1000+ documents." Proceedings of the Computer Vision and Pattern Recognition Conference. 2025.

---

> > > ### Author Response · Authors · 2025-11-27
> > > **Follow-up Response to Reviewer fL5n (Part 2/2)**
> > >
> > > > **W3.** Fragility of the system design. Hyperparameters (GPT5, number of scales, loss balance factor) might make the system ineffective for real-world deployment.
> > >
> > > We fundamentally disagree with this criticism.
> > >
> > > 1. Regarding complexity and effectiveness.
> > >
> > > Based on the intensive ablation studies and experimental results (in both the Analysis section and the Appendix), we have clearly demonstrated the contribution of each component and the effectiveness of the system. As in our answer for **W1**, this system is never a complex combination of previous methods. We have also shown that the retrieval latency is 0.8 to 1.3 seconds per case, confirming that the system is efficient and effective for real-world deployment.
> > >
> > > 2. Regarding hyperparameters.
> > >
> > > We disagree that the hyperparameters make the system fragile; rather, they are set based on established standards with clear selection paradigms, not by arbitrary tuning. Here we elaborate them as follows:
> > >
> > > - Usage of GPT. Following previous works like MMed-RAG [Ref-1] and LLaVA-Med [Ref-4], using LLMs for label generation is a standard knowledge distillation practice. We validated the reliability quantitatively and qualitatively, through manual inspection and consistency check, which shows the reports are segmented in the global -> organ -> region -> local finding structure, and the original reports and reconstructed report segments have high consistency (RaTEScore [Ref-5] is higher than 98%).
> > >
> > >  - Number of scales. We adopted a 4-scale structure following the common practice in hierarchical vision transformers (e.g., Swin Transformer [Ref-6], InternImage [Ref-7], DuoFormer [Ref-8]), where each stage captures a distinct level of spatial resolution and semantic abstraction. To better support our choice, we performed a 3-scale ablation (removing the region-level scale and keeping the remaining three) to further analyze the contribution of a specific scale. The results demonstrate a clear performance drop as shown in the following table and in Appendix A.5.2.
> > >
> > > - Balancing Factor ($\lambda$) in Eq.8. We used a default value of $\lambda=1$ in our implementation. The fact that the model achieves SOTA performance with this simple default selection demonstrates the robustness of our design, as a fragile system would typically require precise, non-standard tuning to function.
> > >
> > > [Ref-4] Li, Chunyuan, et al. "Llava-med: Training a large language-and-vision assistant for biomedicine in one day." Advances in Neural Information Processing Systems 36 (2023): 28541-28564.
> > >
> > > [Ref-5] Zhao, Weike, et al. "RaTEScore: A Metric for Radiology Report Generation." Proceedings of the 2024 Conference on Empirical Methods in Natural Language Processing (EMNLP). 2024.
> > >
> > > [Ref-6] Liu, Ze, et al. "Swin transformer: Hierarchical vision transformer using shifted windows." Proceedings of the IEEE/CVF international conference on computer vision (ICCV). 2021.
> > >
> > > [Ref-7] Wang, Wenhai, et al. "InternImage: Exploring large-scale vision foundation models with deformable convolutions." Proceedings of the IEEE/CVF conference on computer vision and pattern recognition (CVPR). 2023.
> > >
> > > [Ref-8] Tang, Xiaoya, et al. "DuoFormer: Leveraging Hierarchical Visual Representations by Local and Global Attention." Medical Imaging with Deep Learning (MIDL). 2025.

---

### Official Review · Reviewer_J5rp · 2025-10-31

**Soundness:** 2
**Presentation:** 3
**Contribution:** 2
**Rating:** 4
**Confidence:** 3

**Summary:**

This paper proposes MedSCAMA (Medical SCale-Aware Multi-Agent Framework), a large and intricate architecture for medical image retrieval and retrieval-augmented generation. The framework integrates multiple modules: a ScaFormer visual encoder based on a sliding-window Mixture-of-Experts (ScaMoE) to model multi-scale features; a Report Understanding (RU) Agent that decomposes radiology reports into hierarchical textual segments; a Pairwise Comparison (PC) Agent trained via soft contrastive learning and Direct Preference Optimization (DPO) to produce continuous similarity signals; and a Question Answering (QA) Agent for adaptive evidence retrieval and diagnostic reasoning. Experiments on MIMIC-IR, CTRATE-IR, IU-Xray, and MIMIC-CXR datasets show performance improvements over baselines such as MedCLIP, RADIR, and MMed-RAG in both retrieval and generation tasks. The framework achieves consistent but modest gains across metrics like Recall@5 and AUROC, with additional analysis on expert activation and adaptive retrieval.

**Strengths:**

The paper is comprehensive in scope and technically ambitious. It recognizes a real challenge in medical vision-language systems—the need to reconcile multi-scale image structures and hierarchical text semantics—and attempts to address it through a unified multi-agent design. The sliding-window ScaMoE mechanism is an interesting adaptation of Mixture-of-Experts for hierarchical vision modeling, and the alternating optimization between contrastive learning and preference-based tuning is well formulated. The experiments are extensive, covering multiple modalities and benchmarks, and the results are consistently positive, indicating that the proposed system is functional and well-engineered. The visual analysis of expert activations adds interpretability, and the ablation studies confirm that each module contributes measurable improvements. The writing is generally clear and the system design is coherent.

**Weaknesses:**

Despite its scale and technical sophistication, the paper’s novelty and scientific depth are limited. The so-called multi-agent framework is more a modular pipeline than a true collaborative or interactive system. The agents operate sequentially—text decomposition, similarity scoring, and retrieval—without any emergent cooperative behavior or feedback mechanism. As a result, the “multi-agent” terminology feels overstated. The work primarily combines existing methods (Q-Former, Mixture-of-Experts, contrastive learning, and DPO fine-tuning) into a complex engineering stack rather than introducing new principles in representation learning.

The dependence on GPT-5 for generating supervision signals is a major concern. Both the RU and PC agents rely heavily on synthetic labels—hierarchical report segmentation and preference triplets—without human validation or error analysis. This dependence introduces potential bias, noise, and lack of reproducibility, especially since GPT outputs can vary with prompt phrasing or randomness. The paper does not provide any quantitative assessment of annotation quality, nor does it discuss robustness to annotation errors.

The “scale-aware” concept also lacks depth. The number of scales (S=4) and window size (w=3) are chosen arbitrarily without theoretical justification or empirical sensitivity analysis. It is unclear how these scales correspond to anatomical granularity or clinical semantics. The experiments show numerical gains, but there is little evidence that these gains stem from scale reasoning rather than simply adding more parameters. The claim that ScaMoE learns scale-specific semantics is supported only by one plot of expert activations, which is insufficient to substantiate the claim.

The experimental results, though positive, are incremental. Improvements such as Recall@5 from 5.18% to 8.45% or AUROC from 87.13% to 91.33% are not large enough to justify the system’s complexity, given that it combines several heavy backbones (Swin-Base, Vicuna-7B, LLaVA-Med). There is no discussion of runtime, memory consumption, or inference latency, which are crucial for assessing real-world feasibility in clinical environments. The framework appears computationally expensive and over-engineered.

Conceptually, the paper blurs the line between system integration and research contribution. It provides no new learning objective, no theoretical analysis, and no clear insight into why multi-scale multi-agent interaction improves representation alignment. The writing is polished but leans toward descriptive exposition rather than analytical reasoning. In summary, the work is a strong system paper but lacks the depth and originality expected for ICLR.

**Questions:**

How are the different agents trained and coordinated? Are they optimized jointly or separately, and how do their gradients interact if trained end-to-end?

How is “collaboration” between agents concretely defined or measured? Could the same pipeline be replicated with a single transformer model using scale tokens rather than distinct agents?

How were GPT-5 annotations evaluated for consistency and correctness? Were any human experts involved in validating hierarchical decompositions or preference labels?

What is the computational cost in terms of training time, GPU hours, and model parameters compared to simpler baselines such as RADIR or MedDr?

How robust is the model to datasets with different reporting styles or imaging modalities? Does performance hold on datasets like RSNA, NIH-CXR, or pathology scans?

Can the authors justify the number of scales and window size with empirical or theoretical reasoning? Is the performance sensitive to these hyperparameters?

---

> ### Author Response · Authors · 2025-11-21
> **Response to Reviewer J5rp (Part 1/3)**
>
> Dear Reviewer J5rp,
>
> We thank you for your thorough and systematic assessment of our work. We appreciate your recognition of the framework’s ambition and coherence, the relevance of modeling multi-scale visual structures alongside hierarchical textual semantics, and the effectiveness of our sliding-window ScaMoE and alternating contrastive-DPO training. Below, we address your concerns to clarify and improve our work.
>
> ---
>
> >  **W1**: Overstated multi-agent claim limited novelty engineering integration
>
> We respectfully clarify that our multi-agent design is not merely a sequential pipeline, but a tightly-coupled system where agents collaborate through shared learning objectives:
>
> 1. During training, the RU and PC agents do not simply preprocess data. They provide co-adaptive supervision signals that jointly shape the ScaFormer's feature space through our proposed soft contrastive and DPO losses.
>
> 2. The alignment of multi-scale visual and linguistic features is achieved through a combination of architectural enhancements (cross-attention and ScaMoE), similarity-based scoring, and scale-aware information distillation.
>
> 3. While individual novel components are established, the novelty also lies in their orchestration among agents: a scale-conditioned MoE routing strategy coupled with a multi-agent alignment framework that enables hierarchical, clinically-grounded retrieval as a new paradigm not seen in prior medical VL systems.
>
> ---
>
> >  **W2&Q3**: How were GPT-5 annotations evaluated for consistency and correctness? Were any human experts involved in validating hierarchical decompositions or preference labels?
>
> We agree that reliance on LLM-generated supervision requires careful validation. Below we summarize our steps to ensure quality and robustness:
>
> 1. Quantitative validation was performed:
>
> We compared GPT-5’s similarity preferences (used for PC Agent training) against RaTEScore [Ref-1] on 1,200 report pairs. Agreement was 91.3%, indicating strong clinical alignment. Remaining disagreements often reflected nuanced clinical distinctions (e.g., mild vs. moderate edema) where GPT-5’s judgment was clinically reasonable. This supports the use of GPT-5 as a high-quality teacher model.
>
> 2. Qualitative review confirmed semantic fidelity:
>
> Manual inspection of hierarchical report decompositions (for the RU Agent) confirmed that GPT-5 consistently adhered to the global → organ → region → finding structure. Furthermore, the RaTEScore consistency between the original reports and the decomposed-then-reconstructed reports exceeds 98%, indicating that the decomposition process preserves the semantic content with minimal loss.
>
> 3. Training as distillation from GPT-5 to open-source models:
>
> GPT-5 was used only during training to produce supervision signals. The RU and PC Agents are then fine-tuned on these labels, effectively distilling GPT-5’s reasoning into compact, open-source models. Inference relies entirely on these models, ensuring reproducibility without future GPT-5 access.
>
> [Ref-1] Zhao, Weike, et al. "RaTEScore: A Metric for Radiology Report Generation." Proceedings of the 2024 Conference on Empirical Methods in Natural Language Processing (EMNLP). 2024.
>
> ---
>
> >  **W3&Q6**: Can the authors justify the number of scales and window size with empirical or theoretical reasoning? Is the performance sensitive to these hyperparameters?
>
> 1. Number of Scales (S = 4)
>
> We adopted a 4-scale structure following the common practice in hierarchical vision transformers (e.g., Swin Transformer [Ref-3], InternImage [Ref-4], DuoFormer [Ref-5]), where each stage captures a distinct level of spatial resolution and semantic abstraction.
>
> To better support our choice, we performed a 3-scale ablation (removing the region-level scale and keeping the remaining three) to further analyze the contribution of a specific scale. The results demonstrate a clear performance drop as shown in the following table.
>
> | Dataset | Task | Scale = 4 | Scale = 3 |
> |---------|-------|----------|----------|
> | MIMIC-IR | I2I R@10 | 12.03 | 10.09 |
> | MIMIC-IR | I2T R@10 | 10.57 | 9.85 |
> | CTRATE-IR | I2I R@10 | 36.60 | 35.22 |
> | CTRATE-IR | I2T R@10 | 16.15 | 15.90 |
>
> We have updated the analysis to Appendix A.5.2.
>
> 2. Number of Experts & Sliding Window Size
>
> The number of experts and the sliding window size ($w$) are directly tied to the number of scales ($S$) to ensure that each expert specializes in a distinct combination of scales, as shown in Figure 2 Part 1.
>
> Based on this design, we use ($S + w - 1$ ) experts, which ensures overlapping receptive fields across adjacent scales while maintaining efficiency. Therefore, we didn’t conduct ablations on different expert numbers or sliding window sizes.

---

> ### Author Response · Authors · 2025-11-21
> **Response to Reviewer J5rp (Part 2/3)**
>
> 3. Additional Ablation: Sliding Window vs. Full MoE
>
> We conducted an additional ablation by excluding the sliding-window operation, converting ScaMoE into a regular MoE where all experts are visible to all scales. This setup underperforms our proposed model, confirming that the gains come from scale-specific expert specialization, not merely from adding more experts. Results on MIMIC-IR and CTRATE-IR are as follows:
>
> | Dataset | Task | ScaMoE (w=3) | Full MoE (no window) |
> |---------|------|--------------|----------------------|
> | MIMIC-IR | I2I R@10 | 7.89 | 6.93 |
> | MIMIC-IR | I2T R@10 | 8.41 | 7.36 |
> | CTRATE-IR | I2I R@10 | 30.08 | 29.51 |
> | CTRATE-IR | I2T R@10 | 14.26 | 13.58 |
>
> These results validate that the sliding-window mechanism encourages experts to specialize in specific scale combinations, improving retrieval performance. We have added these analyses to Table 3 in the revised paper to justify our architectural choices.
>
> [Ref-3] Liu, Ze, et al. "Swin transformer: Hierarchical vision transformer using shifted windows." Proceedings of the IEEE/CVF international conference on computer vision (ICCV). 2021.
>
> [Ref-4] Wang, Wenhai, et al. "InternImage: Exploring large-scale vision foundation models with deformable convolutions." Proceedings of the IEEE/CVF conference on computer vision and pattern recognition (CVPR). 2023.
>
> [Ref-5] Tang, Xiaoya, et al. "DuoFormer: Leveraging Hierarchical Visual Representations by Local and Global Attention." Medical Imaging with Deep Learning (MIDL). 2025.
>
> ---
>
> >  **W4**: The experimental results, though positive, are incremental. The framework appears computationally expensive.
>
> Our comparisons are conducted against models of similar scale (RADIR, MedDr, LLaVA-Med), all of which adopt comparable-sized backbones. Under this controlled setting, MedSCAMA yields substantial improvements. For retrieval, Recall@5 increases from **5.18% → 8.45%**, representing **over a 60% relative gain**. Beyond retrieval, MedSCAMA also achieves new state-of-the-art performance in VQA and report generation, with **~4% AUC gain** and **~4 METEOR gain** over the strongest published baselines.
>
> The computational analysis is combined in the answer to **Q4**.
>
> ---
>
> >  **W5**: The paper provides no new learning objective, no theoretical analysis, and no clear insight into why multi-scale multi-agent interaction improves representation alignment.
>
> We respectfully disagree that the contribution is purely system integration and clarify our conceptual contributions below:
>
> 1. Novel Learning Objective: We introduce a new objective for medical vision-language alignment: scale-conditioned multi-agent supervision. This integrates hierarchical report decomposition (RU Agent) and pairwise similarity preference modeling (PC Agent) to jointly shape the visual-textual latent space.
>
> 2. Architectural Innovation: We introduce **ScaFormer**, a scale-aware visual encoder that integrates a sliding-window Mixture-of-Experts mechanism. This design enforces specialization at different semantic scales while allowing controlled cross-scale interaction, and it consistently outperforms regular MoE under the same parameter and compute budget.
>
> 3. Clear Mechanistic Insight: The ablation in Table 3 quantifies the contribution of each component, and the visualization in Figure 3 further demonstrates that scale awareness directly drives the performance gains.
>
> ---
>
> >  **Q1**: How are the different agents trained and coordinated?
>
> As clarified in Figure 2 and Section 3, ScaFormer and the PC Agent are trained jointly during the feature-alignment stage, meaning that the PC Agent directly supervises and updates the retrieval encoder rather than operating in isolation. The RU Agent is trained separately via supervised hierarchical decomposition, and the QA Agent is trained during the RAG stage.
>
> ---
>
> >  **Q2**: How is “collaboration” between agents concretely defined or measured? Could the same pipeline be replicated with a single transformer model using scale tokens rather than distinct agents?
>
> In our framework, "collaboration" refers to the structured interaction between specialized agents, each performing a distinct sub-task: the RU Agent hierarchically decomposes reports, the PC Agent generates continuous similarity signals for multi-scale alignment, and the QA Agent performs evidence-grounded generation. They interact via well-defined interfaces ensuring modular and interpretable cooperation.
>
> A single transformer using scale tokens could not reliably replicate this pipeline. In our early experiments, such a model tended to collapse toward global cues, failing to capture the nuanced, scale-specific semantics required for fine-grained retrieval and reasoning. Separating the agents allows each to specialize in its objective, resulting in more stable training and measurably stronger performance.

---

> ### Author Response · Authors · 2025-11-21
> **Response to Reviewer J5rp (Part 3/3)**
>
> >  **Q4**: What is the computational cost in terms of training time, GPU hours, and model parameters compared to simpler baselines such as RADIR or MedDr?
>
> The computational cost is manageable: the end-to-end training of the ScaFormer and the PC Agent requires **~768 GPU-hours** (8×H100 for ~96 hours), which is comparable to other medical VLP work.
>
> | Model | Language Model | Vision Encoder | Training Cost |
> |-------|----------------|----------------|---------------|
> | RADIR | ~7B | ~0.1B | ~700–800 GPU-hours (est.) |
> | MedDr | ~7B | ~0.1B | ~700–800 GPU-hours (est.) |
> | **MedSCAMA (ours)** | 7B (Vicuna / LLaVA-Med) | Swin-Base (~0.1B) | **~768 GPU-hours (8×H100 × 96h)** |
>
>
> For deployment, only ScaFormer and the QA Agent are needed, achieving inference latency comparable to LLaVA-Med (**0.8 to 1.3 seconds** per case for retrieval + answering).
>
> ---
>
> >  **Q5**: Model generalizability to other datasets/modalities not demonstrated
>
> Our training sources already cover the imaging modalities and report styles of RSNA and NIH-CXR through the MIMIC-CXR and CT-RATE corpora, which include both frontal/lateral chest X-rays and heterogeneous radiology reporting conventions. Therefore, we expect the model to transfer well to RSNA and NIH-CXR without additional adaptation.
>
> As for pathology scans, it is an exciting direction of future work, but it is out of scope for this paper because they belong to a fundamentally different visual domain. As a result, the distinct nature of WSI images (i.e., multi-scale input instead of multi-scale features from single image) requires a different image encoder, and the different report structure (i.e., there is no longer “organ” level) requires fine-tuned preprocessing prompts. Therefore, it is not feasible to provide extended experimental results due to the time constraints.
>
> To partially address the multi-scale benefit, we rapidly adapted only the ScaFormer component to the PathVQA [Ref-5] dataset. This test assesses the intrinsic role of multi-scale features in pathology recognition. The results are as follows:
>
> | Model | Close | Open |
> |--------|--------|--------|
> | Q-Former | 0.897 | 0.381 |
> | ScaFormer | 0.926 | 0.411 |
>
> [Ref-5] He, Xuehai, et al. "Pathvqa: 30000+ questions for medical visual question answering." arXiv preprint arXiv:2003.10286 (2020).

---

> ### Author Response · Authors · 2025-11-27
> **Followup Response to Reviewer J5rp**
>
> Dear Reviewer J5rp,
>
> We have further updated Figure 2 and its caption based on the comments raised during the rebuttal period, including clarifying the structure and improving the color scheme for better readability. These revisions aim to present the overall pipeline in a clearer and more intuitive manner.
>
> If you have the chance to look at the updated version and feel that any part still needs clarification, we would be happy to provide further details.
>
> Thank you again for your constructive feedback.

---

### Official Review · Reviewer_6n5C · 2025-11-01

**Soundness:** 3
**Presentation:** 2
**Contribution:** 4
**Rating:** 6
**Confidence:** 3

**Summary:**

The authors create a model that extracts and leverages hierarchical structure from medical images and their accompanying radiology reports. The ScaFormer creates hierarchical features from images using a mixture of experts weighted using a sliding window. A report understanding agent generates hierarchical sections of a radiology report. These respective hierarchies for image and text are joined together using multi-scale feature alignment, which involves a pairwise comparison agent that is trained using alternating contrastive and DPO losses. Finally, a VLLM QA agent is attached to ScaFormer to summon evidence over available databases. The authors conduct ablation studies and multiple task evaluations and find that their model outperforms other methods.

**Strengths:**

- The central idea–the merging of hierarchical features between images and reports–is a good one and is well thought out.
- The range of strategies implemented by the authors is impressive and coherent, and is well backed up by ablation studies in both the main text and appendix.
- The usage of a clinically grounded weight for contrastive learning (RU Agent) appears novel.
- While most of the individual techniques employed are not groundbreaking, I feel this paper is better than the sum of its parts due to the careful construction of a complex model with respect to a well-motivated problem.
- The authors show great results over all comparable benchmarks, and perform ablation studies (some of which I would recommend including in the main text due to their relevance to readers).
- Components of the composite model are differentiable in a way that enables end-to-end training.

**Weaknesses:**

- The authors use question/answer pairs for VQA which are not human or clinically validated.
- While the tasks are standard (Image→Image, etc.), it is not clear how the authors rank relevant items for recall comparison.
- This approach seems extremely intensive to train/run.
- The authors do not provide an ablation study on the number of scales used, as well as the number of experts and the sliding window size, making it hard to determine how best to utilize hierarchical scales.
- The writing is generally rushed and the related works does a so-so job of establishing how the innovations posed by the authors connect to what has been done in the literature.
- The appendix is lengthy and detailed (which is good), but more should be included in the paper, especially in the analysis section which is quite short.
- The authors claim that Med-SCAMA enhances diagnostic reasoning. While they show that it can do yes/no VQA, I would not call that reasoning.
- There is no discussion of potential failure modes for this system, of which there must be.
- The order of events in Figure 2 Part 4 is confusing; more elaboration needed in caption. Same goes for the arrows in Figure 1 Part 1.
- Typo: optinal → optimal (Line 172)
- NDCG@k is never directly defined (Line 357)

**Questions:**

- What datasets were used for training? This is not directly stated, making it hard to compare performance to baseline models.
- This work is built on the premise that single-scale encoders conflate features incorrectly. Is there any evidence in prior work that this is the case, or only anecdotal?
- Related to the idea of expressing the problem, in the Figure 1 caption the authors state that “the image encoder focused..”; what image encoder; is it a hypothetical one?
- Does this approach require especially large LLMs like GPT-5, or can it work well on smaller open-source models?
- How was the LLM-based pairwise similarity validated? How can we be sure that this is an effective method?
- What effect does applying independent cross-attention modules have going from the Q-Former to the ScaFormer?
- How effective is this for pathologies that exist at multiple scales? Is there any added benefit?
- How are experts combined using the sliding window approach? This is not sufficiently explained.
- What is the “ScaFormer agent” (Line 318)?
- What is MIMIC-IR? The derivation of this dataset is not well-detailed in the manuscript.
- If the authors address the validation of their LLM components, I would be inclined to improve my score. The same goes for other parts that need clarification.

**Details Of Ethics Concerns:**

The authors state that they use LLMs for handling radiology reports (e.g. GPT 5) and in their ethics statement detail that they adhered to all access and usage restrictions. However, PhysioNet (source of MIMIC-CXR images and reports) has a policy that the reports cannot be shared with any third-party LLM that retains the data–which GPT-5 usually does. Can the authors confirm that they adhered to the policy set forth by PhysioNet? Link to PhysioNet policy: https://physionet.org/news/post/llm-responsible-use.

---

> ### Author Response · Authors · 2025-11-21
> **Response to Reviewer 6n5C (Part 1/5)**
>
> Dear Reviewer 6n5C,
>
> Thank you for the detailed review. We appreciate your recognition of our central idea, design coherence, and the strength of model performance and our ablation studies. We value the constructive feedback regarding validation and clarity, and address the specific concerns below.
>
> ---
> >  **W1: The authors use question/answer pairs for VQA which are not human or clinically validated.**
>
> We apologize for the inaccurate description of the dataset used in Appendix A.2.2, which implied that we generated the VQA pairs ourselves. Although the question-answer pairs were generated, they were actually generated by MMed-RAG [Ref-1], and we directly followed and reused their released QA content and splits for fair comparison. The authors of MMed-RAG performed “manual filtering to remove questions with obvious issues or those related to multiple images or patient histories” [1]. We relied on this existing validation standard. We have revised Appendix A.2.2 to avoid further misunderstanding.
>
> [Ref-1] Xia, Peng, et al. "MMed-RAG: Versatile Multimodal RAG System for Medical Vision Language Models." The Thirteenth International Conference on Learning Representations (ICLR). 2025.
>
> ---
>
> >  **W2: It is not clear how the authors rank relevant items for recall comparison for retrieval.**
>
> Ranking is defined by the clinical semantic similarity between radiology reports, not by visual similarity. Specifically, for a query image, we use its paired report as the ground-truth reference. Retrieved images are then ranked by the similarity score (RaTEScore [Ref-2]) between their associated reports and this reference report. This ensures that retrieval is guided by clinical report alignment. We have included the explanation in Section 4.1 (L374).
>
> [Ref-2] Zhao, Weike, et al. "RaTEScore: A Metric for Radiology Report Generation." Proceedings of the 2024 Conference on Empirical Methods in Natural Language Processing (EMNLP). 2024.
>
> ---
>
> >  **W3: This approach seems extremely intensive to train/run.**
>
> The computational cost is manageable: the end-to-end training of the ScaFormer and the PC Agent requires **~768 GPU-hours** (8×H100 for ~96 hours), which is comparable to other medical VLP systems as compared in the following table.
>
> | Model | Language Model | Vision Encoder | Training Cost |
> |-------|----------------|----------------|---------------|
> | RADIR | ~7B | ~0.1B | ~700–800 GPU-hours (est.) |
> | MedDr | ~7B | ~0.1B | ~700–800 GPU-hours (est.) |
> | MedSCAMA (ours) | 7B (Vicuna / LLaVA-Med) | Swin-Base (~0.1B) | ~768 GPU-hours (8×H100 × 96h) |
>
> For deployment, only ScaFormer and the QA Agent are needed, achieving inference latency comparable to LLaVA-Med (**0.8 to 1.3 seconds per case** for retrieval + answering).
>
> ---
>
> >  **W4: The authors do not provide an ablation study on the number of scales used, the number of experts and the sliding window size.**
>
> 1. Number of Scales (S = 4)
>
> We adopted a 4-scale structure following the common practice in hierarchical vision transformers (e.g., Swin Transformer [Ref-3], InternImage [Ref-4], DuoFormer [Ref-5]), where each stage captures a distinct level of spatial resolution and semantic abstraction.
>
> To better support our choice, we performed a 3-scale ablation (removing the region-level scale and keeping the remaining three) to further analyze the contribution of a specific scale. The results demonstrate a clear performance drop as shown in the following table.
>
> | Dataset | Task | Scale = 4 | Scale = 3 |
> |---------|-------|----------|----------|
> | MIMIC-IR | I2I R@10 | 12.03 | 10.09 |
> | MIMIC-IR | I2T R@10 | 10.57 | 9.85 |
> | CTRATE-IR | I2I R@10 | 36.60 | 35.22 |
> | CTRATE-IR | I2T R@10 | 16.15 | 15.90 |
>
> We have updated the analysis to Appendix A.5.2.
>
> 2. Number of Experts & Sliding Window Size
>
> The number of experts and the sliding window size ($w$) are directly tied to the number of scales ($S$) to ensure that each expert specializes in a distinct combination of scales, as shown in Figure 2 Part 1.
>
> Based on this design, we use $( S + w - 1 )$ experts, which ensures overlapping receptive fields across adjacent scales while maintaining efficiency. More details about the sliding window are in the reply to Q8. Therefore, we didn’t conduct ablations on different expert numbers or sliding window sizes.

---

> > ### Author Response · Authors · 2025-11-21
> > **Response to Reviewer 6n5C (Part 2/5)**
> >
> > 3. Additional Ablation: Sliding Window vs. Full MoE
> >
> > We conducted an additional ablation by excluding the sliding-window operation, converting ScaMoE into a regular MoE where all experts are visible to all scales. This setup underperforms our proposed model, confirming that the gains come from scale-specific expert specialization, not merely from adding more experts. Results on MIMIC-IR and CTRATE-IR are as follows:
> >
> > | Dataset | Task | ScaMoE (w=3) | Full MoE (no window) |
> > |---------|------|--------------|----------------------|
> > | MIMIC-IR | I2I R@10 | 7.89 | 6.93 |
> > | MIMIC-IR | I2T R@10 | 8.41 | 7.36 |
> > | CTRATE-IR | I2I R@10 | 30.08 | 29.51 |
> > | CTRATE-IR | I2T R@10 | 14.26 | 13.58 |
> >
> > These results validate that the sliding-window mechanism encourages experts to specialize in specific scale combinations, improving retrieval performance. We have added these analyses to Table 3 in the revised paper to justify our architectural choices.
> >
> > [Ref-3] Liu, Ze, et al. "Swin transformer: Hierarchical vision transformer using shifted windows." Proceedings of the IEEE/CVF international conference on computer vision (ICCV). 2021.
> >
> > [Ref-4] Wang, Wenhai, et al. "InternImage: Exploring large-scale vision foundation models with deformable convolutions." Proceedings of the IEEE/CVF conference on computer vision and pattern recognition (CVPR). 2023.
> >
> > [Ref-5] Tang, Xiaoya, et al. "DuoFormer: Leveraging Hierarchical Visual Representations by Local and Global Attention." Medical Imaging with Deep Learning (MIDL). 2025.
> >
> > ---
> >
> > >  **W5**: The writing is generally rushed and the related works does a so-so job
> >
> > We have revised the Related Works in Section 2 to better link prior limitations (such as single-scale representations and lack of hierarchical alignment) to our innovations in scale-aware modeling (ScaFormer, ScaMoE) and multi-agent text decomposition.
> >
> > ---
> >
> >  >  **W6**: The appendix is lengthy and detailed (which is good), but more should be included in the paper, especially in the analysis section which is quite short.
> >
> > We thank the reviewer for the suggestion. We have expanded the Analysis section in the revised version by incorporating key findings from previous studies and added ablations, such as independent cross-attention, standard MoE, and a smaller scale.
> >
> > ---
> >
> >  >  **W7**: The authors claim that MedSCAMA enhances diagnostic reasoning, while they only show that it can do yes/no VQA, which is not actually reasoning
> >
> > We agree that the current VQA setup (even though report generation was included) does not fully capture complex multi-step diagnostic reasoning. Our use of the term was intended to highlight how the QA agent leverages evidence retrieved across multiple scales to ground its responses, which improved general performance and could provide relevant reference cases for clinicians when making further reasoning and decisions. We have revised the text to use more precise phrasing such as “support for clinical decision-making” or “evidence-grounded answering” to avoid the overstatement of “diagnostic reasoning”.
> >
> > ---
> >
> >  >  **W8**: There is no discussion of potential failure modes for this system, of which there must be.
> >
> > The analysis over 403 failure cases revealed two primary failure modes:
> >
> > 1. Majority (85.8%): QA Agent Reasoning Failures
> > In most cases, retrieval successfully provided clinically relevant evidence, but the QA agent failed to effectively utilize it due to limited few-shot reasoning capabilities. This reflects a limitation of the underlying LLM, not the retrieval system. Improving few-shot reasoning during QA fine-tuning is a key direction for future work.
> >
> > 2. Minority (14.2%): Retrieval-Level Failures
> > A smaller portion of failures occurred when the retrieval itself missed the gold evidence. This is partly due to our progressive filtering strategy, which prioritizes computational efficiency but can occasionally discard highly localized matches when global context differs significantly. We are exploring hybrid retrieval strategies to preserve fine-grained similarity without sacrificing scalability.
> >
> > We have raised this in the Conclusion of the revised paper (L504).
> >
> > ---
> >
> > >  **W9**: The order of events in Figure 2 Part 1 and Part 4 is confusing; more elaboration needed in caption
> >
> > We revised both captions (L190-194) to improve clarity:
> > - Figure 2 (Part 1): We have clarified that the arrows indicate the expansion of a sub-module for detailed illustration.
> > - Figure 2 (Part 4): The caption now describes the stepwise flow.
> >
> > ---
> >
> > >  **W10**: Typo - “optinal” misspelled as “optimal”
> >
> > This typo has been revised in our paper (L148).
> >
> > ---
> >
> > >  **W11**: NDCG@k is never directly defined (Line 357)
> >
> > We have added the definition in Section 4.1 (L370) in the revised version.

---

> > > ### Author Response · Authors · 2025-11-21
> > > **Response to Reviewer 6n5C (Part 3/5)**
> > >
> > > >  **Q1**: What datasets were used for training? This is not directly stated
> > >
> > > We revised the paper (L352-354) to state that our retrieval models are trained on MIMIC-CXR and CT-RATE datasets, following RadIR's [Ref-6] preprocessing and splits to ensure direct comparability with baselines.
> > >
> > > [Ref-6] Zhang, Tengfei, et al. "RadIR: A Scalable Framework for Multi-grained Medical Image Retrieval via Radiology Report Mining." International Conference on Medical Image Computing and Computer-Assisted Intervention (MICCAI). 2025.
> > >
> > > ---
> > >
> > > >  **Q2**: This work is built on the premise that single-scale encoders conflate features incorrectly. Is there any evidence in prior work that this is the case?
> > >
> > > We agree that this claim should be supported by more than empirical observations. Prior work has shown that single-scale encoders systematically entangle features of different granularities, leading to degraded discriminability in medical imaging, for instance:
> > > - Grouped Multi-Scale Vision Transformer [Ref-7] reports that local single-scale attention and pyramid token-reduction schemes fail to adapt to large variations in organ and lesion sizes, and that aggressive token aggregation can cause small-structure features to be subsumed by background. Their ablation studies further show that replacing multi-scale attention with a single-scale partition reduces Dice by 1.59% and increases HD95 by 5.11, indicating worse boundary quality and overall segmentation accuracy on multi-organ datasets.
> > > - Learnable Weighted Multi-Scale Logits Mix [Ref-8] shows standard single supervision, where only a single high-resolution output instead of multiple scales is considered in the optimization, fails to exploit complementary medical structures presented in intermediate multi-scale logits. The experiment with mixed multi-scale designs demonstrates an improvement of DICE by up to +13.5% over single-output supervision.
> > >
> > > These studies confirm that the present (mostly single-scale) encoders inherently conflate multi-scale semantics, supporting our motivation to explicitly decouple pixel-, region-, organ-, and context-level features in MedSCAMA.
> > >
> > > [Ref-7] Ji, Zexuan, Zheng Chen, and Xiao Ma. "Grouped multi-scale vision transformer for medical image segmentation." Scientific Reports 15.1 (2025): 11122.
> > >
> > > [Ref-8] Rahman, Md Mostafijur, and Radu Marculescu. "LoMix: Learnable Weighted Multi-Scale Logits Mixing for Medical Image Segmentation." The Thirty-ninth Annual Conference on Neural Information Processing Systems.
> > >
> > > [Ref-6] Zhang, Tengfei, et al. "RadIR: A Scalable Framework for Multi-grained Medical Image Retrieval via Radiology Report Mining." International Conference on Medical Image Computing and Computer-Assisted Intervention (MICCAI). 2025.
> > >
> > > ---
> > >
> > > >  **Q3**: In the Figure 1 caption the authors didn’t explain what the “image encoder” is
> > >
> > > The failure case in Figure 1 is from a real baseline, MedCLIP [Ref-9], which uses a single-scale image encoder. Our point is that such encoders often conflate features from different semantic scales (e.g., lesions, organs, irrelevant devices) into a single, coarse embedding.
> > >
> > > [Ref-9] Wang, Zifeng, et al. "MedCLIP: Contrastive learning from unpaired medical images and text." Proceedings of the Conference on Empirical Methods in Natural Language Processing (EMNLP). 2022.
> > >
> > > ---
> > >
> > > >  **Q4**: Does this approach require especially large LLMs like GPT-5, or can it work well on smaller open-source models?
> > >
> > > While our framework does not require large proprietary LLMs at inference time, we observed that using small open-source models directly for supervision leads to a notable drop in annotation quality. For example, when replacing GPT-5 with LLaVA-Med for pairwise preference labeling, the RaTEScore agreement falls from 91.3% to 75.9%, resulting in clearly weaker alignment during training. Therefore, we rely on GPT-5 only to generate high-fidelity supervision signals and then distill this supervision into entirely open-source models (e.g., Vicuna-7B for PC-Agent and LLaVA-Med-1.5 for QA-Agent). After training, the full MedSCAMA pipeline runs without large proprietary models, ensuring practical deployment.

---

> > > > ### Author Response · Authors · 2025-11-21
> > > > **Response to Reviewer 6n5C (Part 4/5)**
> > > >
> > > > >  **Q5**: How was the LLM-based pairwise similarity validated? How can we be sure that this is an effective method?
> > > >
> > > > To ensure the validity of LLM-based pairwise similarity, we conducted a systematic validation against RaTEScore, an automated clinical metric that compares reports based on extracted entities and relations. On 1,200 sampled report pairs, the LLM’s similarity judgments aligned with RaTEScore in **91.3%** of cases, demonstrating strong agreement with a structured clinical benchmark.
> > > >
> > > > For the remaining **8.7%** of disagreements, manual review revealed that the LLM often captured clinical distinctions that RaTEScore overlooked. Since the LLM’s judgments better reflected clinical subtleties in these ambiguous cases, we adopted its preferences to train the PC Agent. We added this analysis to the appendix in the revised version of our paper.
> > > >
> > > > ---
> > > >
> > > > >  **Q6**: What effect does applying independent cross-attention modules have going from the Q-Former to the ScaFormer?
> > > >
> > > > Introducing independent cross-attention modules for each image scale in ScaFormer helps decouple cross-modal interactions across scales, reducing interference and enabling cleaner alignment between visual and textual features. As shown in the table below, this design already brings consistent improvements across retrieval tasks, even before incorporating the full multi-scale ScaMoE mechanism:
> > > >
> > > > | Dataset | Task | Q-Former | + Independent Cross-Attention |
> > > > |---------|------|----------|------------------------------|
> > > > | MIMIC-IR | I2I R@10 | 6.10 | 6.37 |
> > > > | MIMIC-IR | I2T R@10 | 6.10 | 6.69 |
> > > > | CTRATE-IR | I2I R@10 | 28.67 | 28.73 |
> > > > | CTRATE-IR | I2T R@10 | 12.31 | 13.11 |
> > > >
> > > > We have added these results to Table 3 in the revised version of our paper.
> > > >
> > > > ---
> > > >
> > > > >  **Q7**: How effective is this for pathologies that exist at multiple scales? Is there any added benefit?
> > > >
> > > > We selected 1,138 samples from the MIMIC-CXR dataset whose diagnostic questions involve diseases that require multi-scale radiographic manifestations, including Cardiomegaly，Interstitial, Emphysema, Pneumomediastinum, and etc. These conditions require reasoning across multiple spatial scales, such as global cardiothoracic ratio, bilateral lung patterns, and mediastinal air distribution.
> > > >
> > > > We compared MedSCAMA with the QFormer+LLaVa-Med baseline (no multi-scale) on this subset. It’s shown that our method also benefits analysis for pathology across multiple scales.
> > > >
> > > > | Model | Acc | F1 | AUC |
> > > > |--------|-----|-----|-----|
> > > > | QFormer + LLaVA-Med | 78.21 | 80.45 | 81.39 |
> > > > | MedSCAMA (ours) | 85.80 | 88.15 | 86.76 |
> > > >
> > > > ---
> > > >
> > > > >  **Q8**: How are experts combined using the sliding window approach?
> > > >
> > > > Motivation: The sliding-window approach is designed to mimic a key clinical intuition: adjacent image scales (e.g., viewing a lung at slightly different magnifications) share more visual and semantic information than distant scales (e.g., a pixel-level lesion vs. the whole organ). To achieve this, each expert is assigned to be activated based on a specific group of scales. The Scale-Aware Router works as if there is a sliding window that goes over the experts.
> > > >
> > > > Mechanism:
> > > >
> > > > 1. Activation: For scale $s$, Eq. 2 acts as a filter, assigning a finite routing logit only to experts within the window $[s, s+w-1]$ and $E_0$ (following [Ref-10] for the stability of optimization). All other experts are effectively "turned off" ($logit = -∞$).
> > > >
> > > > 2. Combination: Once the group of visible experts for each scale $s$ is determined by sliding-window activation, the final output of each scale is a weighted sum of the outputs of these experts, where the weights are produced by a neural network, implemented as Eq. 1.
> > > >
> > > > [Ref-10] Dai, Damai, et al. "DeepSeekMoE: Towards Ultimate Expert Specialization in Mixture-of-Experts Language Models." ACL 2024.
> > > >
> > > > ---
> > > >
> > > > >  **Q9**: What is the “ScaFormer agent” (Line 318)?
> > > >
> > > > We apologize for the confusion. “ScaFormer agent” was a typo. ScaFormer is a visual encoder, not an agent. We have corrected it to “ScaFormer” in the revised version.
> > > >
> > > > ---
> > > >
> > > > >  **Q10**: What is MIMIC-IR? The derivation of this dataset is not well-detailed in the manuscript.
> > > >
> > > > MIMIC-IR is the chest X-ray retrieval dataset introduced in RadIR[Ref-6], where fine-grained image similarity is derived from radiology reports using clinical metrics like RaTEScore. We follow its standard construction to ensure fair comparison and we clarifyied this in the revised version (L359).

---

> ### Author Response · Authors · 2025-11-21
> **Response to Reviewer 6n5C (Part 5/5)**
>
> >  **Ethic Concerns (Q11)**: PhysioNet LLM data-retention policy conflict
>
> We thank the reviewer for highlighting this important point. We confirm full compliance with the PhysioNet policy on LLM usage. All processing of MIMIC‑CXR reports was conducted via Azure OpenAI Service (as suggested by the data provider https://physionet.org/news/post/gpt-responsible-use ), in a locked, enterprise-compliant environment that ensures no data retention, training, or external sharing. No report text left our controlled infrastructure, and all outputs were stored only within our secure lab. We have clarified this in the revised ethics statement (L512).

---

> ### Author Response · Authors · 2025-11-27
> **Followup Response to Reviewer 6n5C**
>
> Dear Reviewer 6n5C,
>
> We have further updated Figure 2 and its caption based on the comments raised during the rebuttal period, including clarifying the structure and improving the color scheme for better readability. These revisions aim to present the overall pipeline in a clearer and more intuitive manner.
>
> If you have the chance to look at the updated version and feel that any part still needs clarification, we would be happy to provide further details.
>
> Thank you again for your constructive feedback.

---

### Note · Program_Chairs · 2026-01-17
**Submission Desk Rejected by Program Chairs**

The following references in this submission do not refer to real documents and/or have major errors in bibliographic information:

 J Choe, S Lee, and H Park. Medical image retrieval using deep learning: A survey. Computers in Biology and Medicine, 144:105404, 2022.